# Simulating the ghost: quantum dynamics of the solvated electron

Jingang Lan [1✉], Venkat Kapil[2], Piero Gasparotto[3], Michele Ceriotti [2], Marcella Iannuzzi [1] & Vladimir V. Rybkin [1✉]

The nature of the bulk hydrated electron has been a challenge for both experiment and theory due to its short lifetime and high reactivity, and the need for a high-level of electronic structure theory to achieve predictive accuracy. The lack of a classical atomistic structural formula makes it exceedingly difficult to model the solvated electron using conventional empirical force fields, which describe the system in terms of interactions between point particles associated with atomic nuclei. Here we overcome this problem using a machine-learning model, that is sufficiently flexible to describe the effect of the excess electron on the structure of the surrounding water, without including the electron in the model explicitly. The resulting potential is not only able to reproduce the stable cavity structure but also recovers the correct localization dynamics that follow the injection of an electron in neat water. The machine learning model achieves the accuracy of the state-of-the-art correlated wave function method it is trained on. It is sufficiently inexpensive to afford a full quantum statistical and dynamical description and allows us to achieve accurate determination of the structure, diffusion mechanisms, and vibrational spectroscopy of the solvated electron.

[1] Department of Chemistry, University of Zurich, Zürich, Switzerland. [2] Laboratory of Computational Science and Modelling, Institute of Materials, Ecole Polytechnique Fédérale de Lausanne, Lausanne, Switzerland. [3] Empa, Swiss Federal Laboratories for Materials Science and Technology, Dübendorf, Switzerland. ✉email: jinggang.lan@chem.uzh.ch; vladimir.rybkin@chem.uzh.ch

The solvated electron, e⁻(aq)[1] has been attracting attention of both experimental and theoretical research for more than half a century[2–7]. Understanding its behavior has fundamental implications for electrochemistry, photochemistry, and high-energy chemistry, as well as for biology: the non-equilibrium precursor of e⁻(aq) is responsible for radiation damage to DNA[8]. Despite its apparent simplicity—it is the smallest possible anion as well as the simplest reducing agent in chemistry, and has been regarded as a realization of a "particle in a box" model[2]—capturing the correct physics of the solvated electron is highly non-trivial. Within density functional theory (DFT), the electronic structure method that has been used most often to study the solvated electron and water, standard density functionals suffer from delocalization error[9], preventing accurate modeling of radicals. In fact, neat water is non-trivial for DFT approximations, though a proper choice of functionals may offer satisfactory results compared to high-level electronic structure benchmarks and experimental observables[10,11]. Accurate description of liquid water can be also achieved with many-body quantum chemistry methods, such as second-order Møller-Plesset perturbation theory (MP2)[12], which are, however, exceedingly expensive. Moreover, MP2 provides satisfactory accuracy for the two- and three-body interactions for water[13] and solvated electron in clusters[14], benchmarked against CCSD(T) calculations. Successful implementation of spin-unrestricted-MP2 (UMP2) energies and forces under periodic boundary conditions has enabled accurate molecular dynamics (MD) of the bulk solvated electron[15,16]. Picosecond-scale MP2-based MD of e⁻(aq)[16], unprecedented in computational complexity, provided a crucial argument in favor of the cavity structure and against the broadly discussed non-cavity structure[17].

A comprehensive characterization of the properties of this system, however, requires by far longer times scale. Likewise, simulation of quantum nuclei at this level of electronic structure theory is beyond computational reach, e.g., path-integral (PI)MD is at least one order of magnitude more demanding than classical MD. Yet, a PI study of the dynamics of an excess electron in water clusters has demonstrated that nuclear quantum effects (NQEs) are very important in its relaxation dynamics[18]. In condensed phase, NQEs are sizeable for redox properties and electron attachment[19]. Moreover, NQEs can often sample regions of the configuration space that are classically inaccessible, and introduce qualitative changes in the behavior of aqueous systems[20,21].

In this work, we report long timescale quantum dynamics of the hydrated electron with an MP2-quality description of the interatomic potential. Using forces and energies computed at the MP2 level of theory (for details see Supplementary Methods (MP2 calculations)), we have trained a Behler-Parrinello Neural Network (BPNN)[22] force field. We discuss in the Supplementary Methods (Machine learning) the construction of the training set (that contains configurations from both classical and quantum MD), the details of the potential, and its benchmarking against reference UMP2 energetics. The timescale we achieve (a total of several hundred picoseconds) allowed us to compute converged structural and dynamical properties (vibrational spectra, and diffusion), and quantify the impact of NQEs and isotope effects on bulk hydrated electron.

## Results

When attempting to model this system using a conventional empirical force field, the excess electron must be described explicitly using a one-electron Schrödinger equation[2]. A NN potential, on the other hand, relies on a representation of the system in terms of interatomic correlations, to reproduce the relationship between a configuration and its energy, which is written as a sum of atom-centered contributions. Even though there is no specific term to account for the solvated electron, its presence is encoded in the structural correlations between the surrounding water molecules, and in the stabilizing effect that the "ghost electron" has on the cavity it occupies. Note that, contrary to what is usually the case, our machine-learning potential should not be run for a different system size than it was trained for. This is because the presence of an excess electron is an additional parameter of the reference calculations, and so the potential is effectively trained at a fixed concentration of electrons. Changing system size without further training would be equivalent to introducing additional electrons, which may lead to nonphysical results.

The NN model proves to be remarkably stable: simulations can be initiated by equilibrating the system with a potential trained on neat water, and then switched to the potential trained with an extra e⁻—effectively modeling the injection of the electron. Spin densities ($\rho_{\mathrm{spin}}(r)$) of the "ghost electron", which are not directly available from the BPNN simulations, are then evaluated for selected frames using DFT with a hybrid functional. We performed 30 independent trajectories simulating the injection process, modeling nuclear quantum dynamics using the thermostatted ring-polymer (TRP) MD method[23] (for details, see Supplementary Methods (Computational methods)), and followed the localization process by visualizing the spin density and the band gap. As shown in Fig. 1a, the initial fully delocalized state of the excess electron (see Fig. 1d) undergoes a pre-solvation regime (see Fig. 1e and ends up in a cavity (see Fig. 1g) (see Movie.S1). Most of the trajectories localized the solvated electron and formed a cavity within 1 ps. This localization dynamics featuring the delocalized, the pre-solvated, and the cavity structures consecutively is in agreement with both experiment and previous ab initio MD simulations[6,7,16,24]. Furthermore, the inclusion of NQEs activates a novel diffusion mechanism involving the formation of a twin cavity (Fig. 1f), as we discuss below.

The structure of the cavity occupied by the extra e⁻ involves 4/5 water molecules, each having one OH moiety pointing toward the center of the cavity (Fig. 1g). Spin density distributions reveal the characteristic negative density regions next to H atoms[16,25]. The radial distribution functions (RDFs) of H and O atoms relative to the spin density center (restricted to single-cavity structures, see Supplementery Methods for the definition, and the results for D₂O) provide a more quantitative structural characterization of the cavity, as shown in Fig. 1h, i. The mean e⁻-O coordination number is 4.5, which is similar to the best DFT estimate of 4.7[26]. NQEs broaden the first peaks in both e⁻-O and e⁻-H, due to zero-point vibrational motion, and reduce considerably long-range order, which is evident from the blurring of the second-neighbor peak. Interestingly, RDFs of oxygen from quantum dynamics have non-zero values within 1.5 Å (less than the covalent radius) from the electron's center. This is consistent with the recently proposed fluxational picture of the solvated electron[27]: quantum delocalization toward the center of the cavity allows to reduce the entropic cost of creating a void.

As shown in the SI, the RDFs for heavy water are very similar to those of classical simulations, which is consistent with the observation of near-perfect compensation of competing quantum effects for D₂O[28]. A detailed analysis of the HB network based on a probabilistic definition of molecule's HB state[29] (see Supplementary Discussion (H-bond species populations)) shows that the formation of the cavity is associated with an increase in the number of undercoordinated water molecules. For each cavity, the absolute number of singly-donating molecules in the cell increases by two, and the number of singly accepting molecules increases by three.

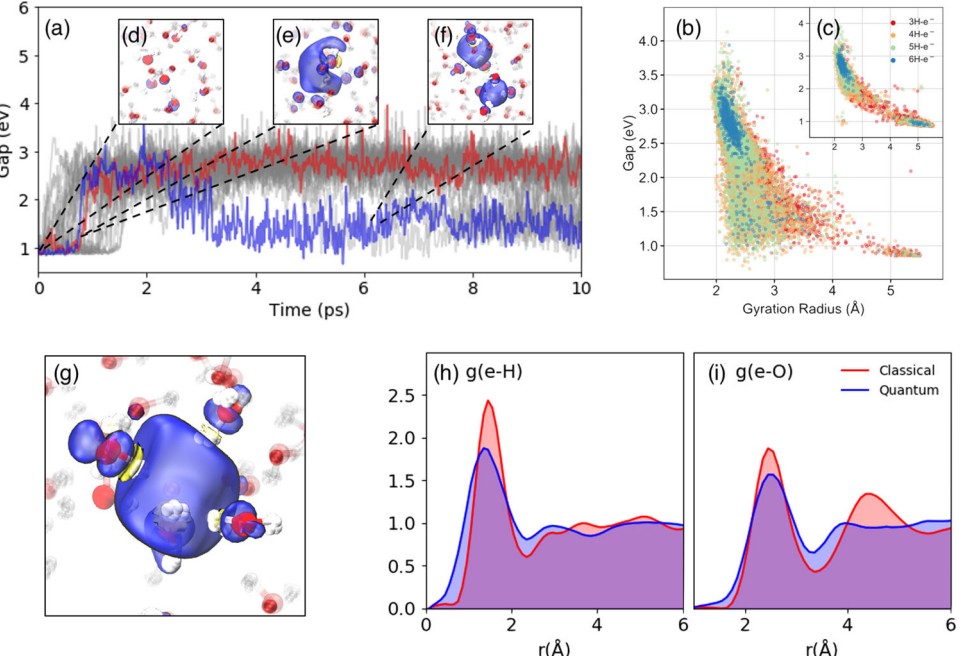

**Fig. 1 Computed properties of the solvated electron. a** Time evolution of the energy gap of the excess electron as obtained from 30 individual trajectories, with two representative curves marked in red (single-cavity) and blue (involving a twin cavity). **b** Band gaps, gyration radii of the spin density distribution with different H-e⁻ coordinations, as obtained from quantum and **c** classical simulations. The coordination number between e⁻ and hydrogen are marked in red (3H-e⁻), orange (4H-e⁻), green (5H-e⁻), and blue (6H-e⁻). Spin densities of the solvated electron from quantum dynamics are shown for **d** the delocalized electron, **e** the pre-solvated electron, **f** a twin cavity, and **g** a single cavity. The yellow isosurface indicates a negative $\rho_{spin}(r)$. **h, i** Radial distribution functions of hydrogen (**h**) and oxygen (**i**) atoms with respect to the center of the solvated electron spin density distribution from classical (red) and quantum (blue) simulations.

The structure of the solvated electron correlates strongly with its electronic properties. Figure 1b shows the joint distribution of the gyration radius and the band gap, color-coded according to the e⁻-H coordination number. For the single cavity, the band gap is inversely correlated to the gyration radius, and directly correlated to the coordination number. The anisotropy of the cavity, including both single and double-cavity structures, is 0.0423 ± 0.0349—highlighting the fact that double-cavity structures are rare and give a small contribution to the average properties of the solvated electron. The most prominent effect of NQEs can be seen in the lower part of Fig. 1b, c. In classical simulations the electron occasionally undergoes delocalization, with a gyration radius >4 Å and a band gap of 1 eV. In quantum simulations, on the other hand, one can observe cavities with a gyration radius between 2 and 3 Å, which are associated with an anomalous low gap (≈1.5 eV) compared to the average gap of ≈2.6 eV. Those states, which are absent in classical simulations, are associated with the twin-cavity state shown in Fig. 1f. This twin cavity is also involved in a transient diffusion process that is only observed in quantum simulations on H₂O, shown in Fig. 2e–h. Quantum simulations on heavy water described below do not reveal this mechanism. The classical, non-transient mechanism is initiated by the formation of the hydrogen bonds (HBs) between a cavity-forming molecule and its unbonded neighboring water molecules (indicated as A and B in Fig. 2a–d). The dangling OH of A, B subsequently binds to the hydrated electron in the cavity, replacing that of a third molecule, C. Finally, molecules B and C form HBs with their neighbors, and unbind from the cavity. This diffusion mechanism is analogous to that responsible for the localization process, which is initiated by a broken HB, which releases a free OH moiety that subsequently acts as a trap[16,24]. The transient diffusion mechanism is characterized by the coexistence of two adjacent cavities, whose

volumes change as the spin density transits from one cavity to the other. The process is initiated by the simultaneous breaking of several HBs next to the cavity (see Fig. 2f, g), forming a second void. The electron can shuttle between the two voids, and may eventually complete a diffusion step (Fig. 2h and Movie.S2). These transient diffusion events are relatively rare but persistent, and are associated with a characteristic structural and electronic signature in Fig. 2f. They do not appear, however, to contribute substantially to the macroscopic diffusion rate. The diffusion coefficient obtained by quantum dynamics (0.40 ± 0.03 Å²/ps) is within the statistical error bar of the classical value (0.36 ± 0.04 Å²/ps), both numbers being in reasonable agreement with the experimental measurements of 0.475 ± 0.048 Å²/ps[30].

The unexpected appearance of the twin-cavity structure calls for additional tests, to verify the reliability of the ML model in a region that was poorly sampled in the reference MP2 calculations, that are mostly associated with classical MD and single-cavity structures. A cross-validation analysis proves that the typical model errors for twin-cavity structures are comparable to that of the original training structures. Furthermore, tests performed with a NN potential trained on an extended set that includes the validation structures exhibit similar behavior to that observed here (see Supplementary Methods (Machine learning)). Thus, the qualitative enhancement of the stability of the double-cavity by NQEs is consistent with the underlying MP2 reference calculations.

In order to understand how an experimental detection of these structures would be possible, we have calculated electronic absorption spectra using time-dependent density functional theory (see Supplementary Discussion (Electronic spectra)). The computed absorption maximum for the single-cavity structures, corresponding to the s-type to p-type orbital transition, is located at ca. 2.2 eV (see Supplementary Figs. 6 and 7) in a reasonable

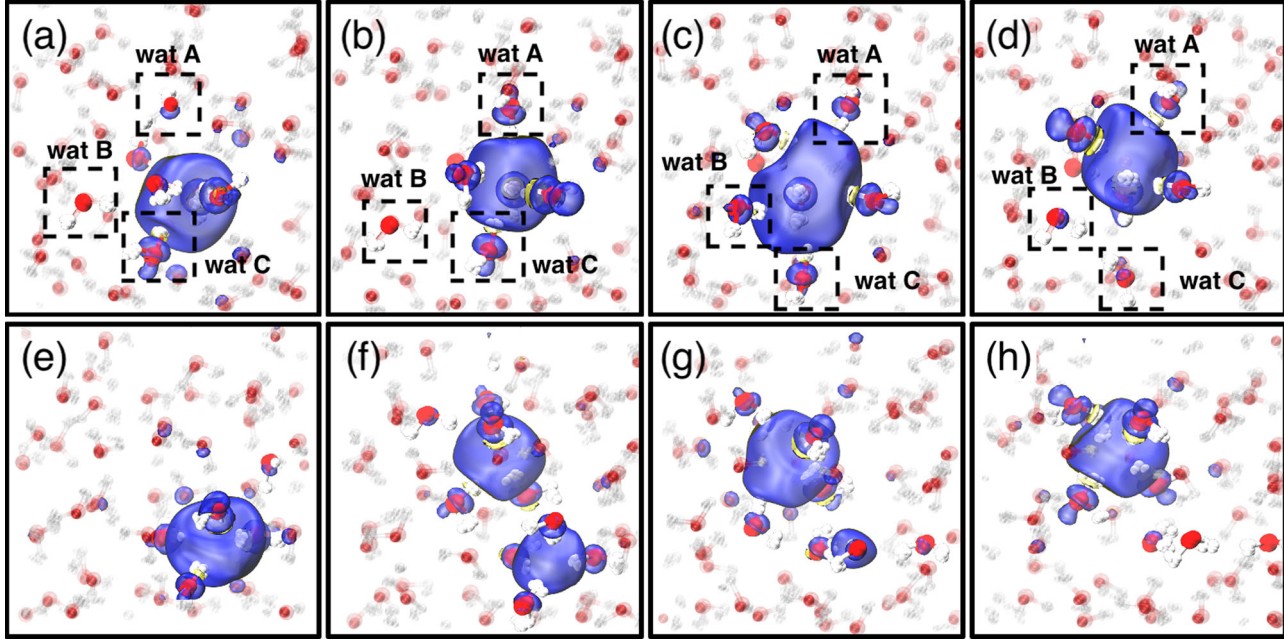

**Fig. 2 Non-transient (top) and transient (bottom) diffusion mechanisms.** Spin densities are shown in blue (positive) and yellow (negative). Non-transient mechanism is initiated by the formation of the hydrogen bonds (HBs) between a cavity-forming molecule and its unbonded neighboring water molecules (indicated as A and B in panels (**a**)–(**d**)). The dangling OH of A, B subsequently binds to the hydrated electron in the cavity, replacing that of a third molecule, C (panel (**c**)). Molecules B and C form HBs with their neighbors, and unbind from the cavity (panel (**d**)). Transient diffusion (panels (**e**) to (**h**)) is characterized by the existence of the double-cavity (panel (**f**)).

agreement with the experiment (1.7 eV[1]). Twin-cavity absorption spectrum is also broad featureless peak with the maximum below 1.0 eV, corresponding to the electron transfer from an s-type orbital localized in one cavity to a similar s-type orbital of the adjacent one. Keeping in mind the low concentration of double-cavity structures (<10%), the contribution of the twin-cavity structures to the total spectrum must be marginal. We expect them to give a signal in transient bleaching experiments[31], although the search should be done at lower frequencies and can be hindered by the method's sensitivity to the low-concentration twin-cavity structures and by the absorption of the excited state[32].

Even though this analysis provides key insights into the static and dynamical behavior of the hydrated electron, it cannot be directly compared with the most commonly used experimental probe, resonance Raman (RR) spectroscopy. RR measurements show a clear signature of the presence of the excess electron as a downshift of the main vibrational peaks relative to those observed in neat water. For the peak associated with the intramolecular bend, one observes downshifts of about 30 cm$^{-1}$ for e$^-$/H$_2$O, and 20 cm$^{-1}$ for e$^-$/D$_2$O[33]. In RR measurements, the vibrational signals involving solvated electrons are enhanced. To disentangle the vibrations of the excess electron's solvation shell, mimicking this effect, we compute the vibrational density of states (VDOS) of the water molecules within 3.5 Å from the spin density distribution center. Our TRPMD results for H$_2$O, D$_2$O, and an artificial system composed of pure HOD, are shown in Fig. 3a–c, with the results from classical MD plotted in panel (a) with a dashed line. Results for the bending peaks are in excellent agreement with experimental values for both light and heavy water, with peak position and the e$^-$-induced shift agreeing with experiments to within 10 cm$^{-1}$. In classical simulations bending peaks are blue-shifted by more than 50 cm$^{-1}$, and almost no e$^-$-induced shift is observed—underscoring the importance of combining accurate electronic structure theory with approximate quantum dynamics of the nuclei.

In the stretching region the agreement is less quantitative; in this region, the best approximate quantum dynamics techniques lead to errors of up to 100 cm$^{-1}$ (although errors from neglecting NQEs are much larger), and the line shape of the VDOS is likely to be a poor proxy of that from RR measurements. However, the red shift induced by the solvated electron (≈150 cm$^{-1}$ for light water) is consistent with that observed in experiments (≈200 cm$^{-1}$)[33]. Semi-quantitative agreement in the downshift is observed also for heavy water. The broadening and shift of the stretching peak are consistent with the increase in defect population observed in the HB network analysis. Specifically, the Raman stretching band of a singly-donating OH is associated with a 200 cm$^{-1}$ red shift[34], and is therefore consistent with the cavity structure of the solvated electron. The RR spectra of isotopically-mixed water (1:2:1 mixture of D$_2$O:HOD:H$_2$O) presents some additional interesting aspects. The spectrum of the solvated electron gives rise to an unusual vibrational feature, containing a doublet in the bending region. This can only arise from HOD molecules close to the cavity: one peak corresponds to those that have the O–H oriented toward the electron, the other to those having the O–D oriented toward it[35]. We predict a peak splitting with two maxima at 1338 and 1399 cm$^{-1}$, which are assigned to H–O–D-e$^-$ and D–O–H-e$^-$ bending modes, which corresponds closely to the experimentally-observed splitting of 60 cm$^{-1}$, although the absolute peak positions are red-shifted.

To avoid the inevitable approximations involved with the modeling of quantum dynamics, one can also probe NQEs by studying equilibrium isotopic segregation at the cavity. The larger downshift of the OH frequency in comparison to the OD frequencies implies that the chemical equilibrium e$^-$(H$_2$O)$_x$(D$_2$O)$_y$(HOD)$_z$ + H$_2$O $\rightleftharpoons$ e$^-$(H$_2$O)$_{x+1}$(D$_2$O)$_{y-1}$(HOD)$_z$ + D$_2$O, is expected to shift to the right. In fact, the equilibrium constant $K_{eq}$ has been experimentally determined to be 1.6 which quantifies the free energy difference associated with the exchange of a D$_2$O molecule with a single H$_2$O in the electron–water complex[33]. Quantum alchemical exchanges calculations, which allow to swap H and D atoms during the

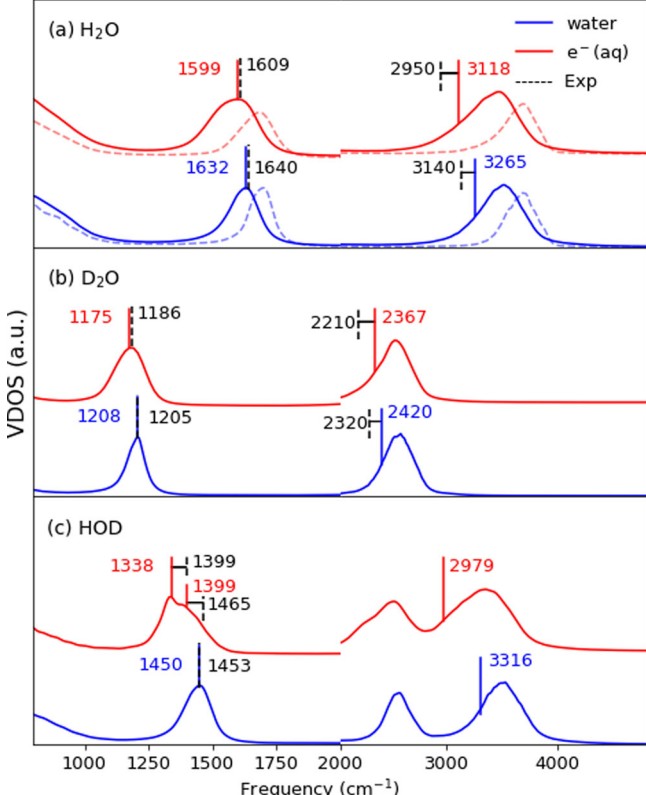

**Fig. 3 Vibrational density of states (VDOS). a** H₂O; **b** D₂O; **c** HOD. Neat water (blue) and of the first coordination shell of a solvated electron (red), computed from the velocity autocorrelation function of atoms from TRPMD, or classical MD (dashed lines in panel (**a**)) simulations. The spectra are averaged over 30 independent 10-ps-long trajectories. The bending mode peak positions and the stretching peak half-maximum position at stretch mode are marked by vertical lines, to facilitate direct comparison with experiments, indicated by black dashed lines[33].

simulations in a Monte Carlo fashion[36], confirm that electron transfer from pure H₂O to D₂O is energetically unfavorable. $K_{eq}$ is estimated to be 1.97 ± 0.24, in good agreement with experiments[33].

By combining state-of-the-art electronic structure methods to compute accurate reference energetics, machine-learning potentials to make thorough statistical sampling feasible, and TRPMD to model the role of quantum nuclear fluctuations on static and time-dependent properties, we provide a comprehensive theoretical study of the behavior of the solvated electron. The accuracy of the model is demonstrated by the quantitative agreement between the computed vibrational density of states and measured resonance Raman spectra, made possible by the uncompromising description of electronic and nuclear quantum behavior of the system. Quantum effects are also clear in the presence of strong isotope effects, that favor the interaction of the solvated electron with ¹H over that with D. By collecting hundreds of picoseconds of quantum dynamics we can describe quantitatively the structure of the cavity that surrounds e⁻(aq), its effect on the HB network, and the dynamics of localization following electron injection, that takes place on a timescale of about 1 ps. Quantum effects enable a previously unknown diffusion mechanism, that is not seen in classical MD and involves two coexisting cavities. The stabilization of extended voids induced by the presence of a solvated electron is yet an unexpected consequence of the interplay of quantum nuclear and electronic effects in water, that might be put to the test by careful isotope substitution experiments, which can use our spectroscopic predictions as guidelines.

## The success of an atom-centered machine-learning scheme in modeling the solvated electron as a "ghost particle" that manifests its presence in the stabilization of the molecules surrounding the associated cavity, without an explicit representation, sets a promising precedent for the modeling of similar charged, defective or excited states, including for instance polarons, holes, excitons, and other quasi-particles.

### Data availability
Data generated and analyzed for this study that are not included in this article and its Supplementary Information are available at https://www.materialscloud.org.

### Code availability
The computer codes used for electronic structure calculations (CP2K[37]), machine learning (n2p2[38]), and in quantum dynamics (i-PI[39]) are described in the Supplementary Methods. The code used for hydrogen bonding analysis (PAMM[40]) is described in Supplementary Discussion/H-bond species populations. Their availability is referenced therein.

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

## Acknowledgements

This work is funded by the Swiss National Science Foundation (SNSF) Sinergia grant and the University Research Priority Program (URPP) for solar light to chemical energy conversion (LightChEC) of the University of Zurich. This work was also supported by a grant from the Swiss National Supercomputing Center (CSCS) under Project ID uzh1, s965. VRR has been supported by the SNSF in the form of Ambizione grant No. PZ00P2_174227. V.K. and M.C. acknowledge support from the NCCR MARVEL, funded by the SNSF. We thank Professor Jürg Hutter for the support and discussions. We thank Joost VandeVondele (CSCS) and Mauro del Ben (Berkeley Lab) for sharing their invaluable MP2 water trajectories. J.L. acknowledges support from a GRC Travel Grant. We are grateful to Prof. J. Herbert (Ohio State University, Columbus) and to Dr. J. Helbing (University of Zurich) for useful discussions.

## Author contributions

J.L., V.K., and V.R. conceived and designed research; J.L., V.K., and V.R. performed research; J.L., V.K., M.C., and V.R. wrote the manuscript; J.L., V.K., P.G., M.C., M.I., and V.R. contributed to the results, analysis, and discussion.

## Competing interests

The authors declare no competing interests.
