## [Peer Review File · Nature Communications]

REVIEWER COMMENTS

Reviewer #1 (Remarks to the Author):

The manuscript by Rybkin and coworkers makes an excellent capstone on a long-running saga to understand the structure and localization dynamics of an “excess” (or solvated) electron in water. It is also a great example of how machine learning has become just another method in the theoretical chemistry toolkit, to be used as needed — and it is put to good use here.

For a time in the 2010s there was some controversy regarding the structure of $e^{-}(aq)$, which had been a question decades ago but had largely been thought a settled issue until the publication of an erroneous Science paper by Schwartz and coworkers in 2010. As pointed out in the present work, condensed-phase MP2 simulations by Rybkin and coworkers (Ref. 13) put the final nail in the proverbial coffin of the non-cavity model, as summarized in Ref. 2, but these simulations are limited to a few picoseconds at tremendous computational expense. Here, the authors have significantly extended the timescale and furthermore facilitated a quantum treatment of the nuclei by using machine learning to convert the condensed-phase MP2 data into an affordable force-field-like representation.

What I like about this work is that it succeeds both as a theoretical methods paper and as a chemical physics paper. On one level, I think this is a great demonstration of the combination of cutting-edge (but exceedingly expensive) ab initio theory, with machine learning techniques, to really do great simulations of a system that is very challenging for theory. At another level, there is a nice confirmation of the localization dynamics (essentially complete within 1 ps) that had been inferred from other simulations and from THz spectroscopy experiments, but is nice to see here from a very high level of theory.

There are a few things that I think should be revised by the authors, largely related to the tone of how certain ab initio methods are described, but these are easily addressed and I wholeheartedly support publication of this work in Nature Communications.

Issues for revision:

(1) I think that the discussion of DFT vs. MP2 for liquid water in the introduction is a bit disingenuous. I understand that a common trope in the introduction to high-profile papers is basically to say “the state of the art has terrible problems”, but that doesn’t mean this is right, and the authors are pushing it a bit. For example, they say “neat water is already challenging for DFT” and cite a Perspective by Michaelides (Ref. 9) titled “How good is DFT for water?”, but if you actually *read* that paper, the answer that Michaelides offers is actually: pretty good. There are some issues, for sure, but the situation is certainly much improved as compared to the early days of ab initio MD and anyway there are issues with the MP2 description of liquid water as well, which are not acknowledged here (Hirata, 10.1021/acs.jpcclett.5b02430; Vandevondele, 10.1021/jz401931f). Specifically, MP2 water is too dense and probably has a too-high boiling temperature. Both issues are likely related to the fact that MP2 significantly overestimates dispersion interactions, and the problems would be much more severe in a liquid where the dispersion effects were more significant. (They are certainly collectively significant in liquid water but amount to only a few percent of the binding energy of water dimer.) The present work can stand on its own without needing to misrepresent the current state-of-the-art. MP2 is a good level of theory (if one can afford it) for this particular system but that doesn’t mean that we should throw out DFT, and it doesn’t mean that MP2 is a good choice for everything even if it is feasible.

(2) Speaking of things that are disingenuous, the authors make comments about how DFT cannot be trusted for radicals due to self-interaction (not wrong but overstated)... and then they proceed to visualize all the data using spin densities computed with hybrid DFT. They use a functional that Ambrosio and Pasquarello have previously used to examine electronic energy levels of both neat liquid water and of e-(aq), and I think this functional is fine for the purpose for which it is used here. That said, it was still rather jarring to segue directly from criticism of DFT in the introduction to repeated use of DFT later in the paper.

(3) On pg. 6, the authors note that their simulated coordination number of 4.5 “corresponds to the experimentally deduced value and is lower than the best DFT estimate of 4.7”. If you put uncertainties on those values representing fluctuations, my guess is they are the same number so I think it’s a cheap shot to say MP2 agrees with experiment but DFT does not. Furthermore, what is the experiment? Those numbers seem about right for various room-temperature simulations but the only experimental work of which I’m aware that gets at coordination number are ESR experiments at liquid nitrogen temperatures, which is not relevant here. It’s possible I’ve missed some literature but it seems like the 4.5 number is likely an indirect inference at best.

(4) Regarding the vibrational spectra, it is compelling that the authors recover a splitting in the HOD bend, which is a feature that Ben Schwartz’s non-cavity model never recovered even as he was pushing the resonance Raman spectrum as the thing that proved the veracity of his non-cavity interpretation. (This cognitive dissonance was noted in Ref. 30.) However, I was surprised by the lack of agreement with experiment in the O-H stretching region, although the simulations do pick up on the broadening on the low-energy side of the OH feature, comparing neat water to e-(aq), which is a hallmark of this system. Do the authors have an explanation for the lack of quantitative agreement in the stretching region? What is the state of anharmonicity in these simulations? The classical calculations I would expect to be overly harmonic and that may account for how this feature appears at higher wavenumbers in the classical simulation, but I don’t have a good understanding of why the errors should be so large once nuclear quantum effects are introduced, at what ought to be a good level of electronic structure theory.

Minor semantic and/or technical issues:

(1) A more standard notation for the density functional that is used would be PBEh(alpha) rather than PBE(alpha), where the “h” indicates that it is a hybrid.

(2) In the simulations of electron injection (Fig. 1a), is the system truly zero-gap at t=0? That would not surprise me, as the electron is probably fully delocalized or at least spread over a number of disjoint pre-existing traps or voids. My question is: how do the authors perform MP2 calculations for a zero-gap system? I realize this is a question mainly about Ref. 13 but I did not review that paper so I am asking it now.

(3) Pg. 7: I do not understand the sentence “These transient diffusion events... are associated with a characteristic structural and electronic signature in Fig. 2(f)”. What are the signatures to which the authors refer?

(4) There are some capitalization issues with the references that probably go back to the BibTeX files the authors are using.

Reviewer #2 (Remarks to the Author):

This manuscript reports quantum Path Integral Molecular Dynamics (PIMD) simulations of the hydrated electron using a Machine Learning (ML) force field with a stated MP2-quality description. The well-tested Behler-Parrinello Neural Network (BPNN) approach is used and trained on prior MP2 Ab Initio Molecular Dynamics (AIMD) simulation data. Interestingly, this results in a force field for the hydrated electron that depends only on the nuclear positions – no explicit description of the excess electron is needed. The picture that results from the path integral simulations is predominantly a cavity description of the hydrated electron, and computed observables agree well with experiment, including localization dynamics, electron radius of gyration, diffusion coefficient & vibrational spectroscopy. Surprisingly, when quantum effects are included, twin cavity structures are occasionally observed (containing half an electron) and these exhibit a novel electron diffusion mechanism.

The manuscript is well written. Furthermore, the new insight into the hydrated electron (if, after addressing my concerns below, is correct) and the methodological development of being able to use atom-based forcefields without explicit inclusion of the excess particle are of significant impact and broad interest; however, my suggestions below have the potential to change the conclusions of the paper and additional analysis I believe is warranted.

Specific criticism/comments/questions (in no particular order):

1) The twin cavity structure observed in PIMD simulations of the hydrated electron is highly surprising. First, this is not merely a rare-event fluctuation: Fig. 1(a) shows many trajectories that visit low band-gap structures characteristic of the twin-cavity state, and they persist for at least 7 ps in at least one trajectory, suggesting the twin cavity is a (meta-)stable state of the hydrated electron in the authors' model. The authors should clarify the percentage of time a twin cavity structure is observed. Nevertheless, I have strong doubts this corresponds to the physical reality of the hydrated electron. Instead, I suspect it is a simulation/ML artefact for the following reasons:

- One would expect the formation of two separate cavities and reduction of charge density in each cavity by half compared to a single cavity to be strongly disfavored in a high dielectric solvent like water. It is hard to imagine any physical driving force to the formation of the twin cavity.
- The BPNN forcefield underlying the PIMD simulations may have issues of overfitting/selection bias that are discussed in more detail in point 2) below.
- The (metastable) twin cavity structure should be detectable in the optical absorption spectrum. The experimental absorption spectrum is a featureless single peak at 1.7 eV.[1] Although the authors did not compute the optical spectrum, they report the average band gap at 2.6 eV (notably higher than the experimental absorption spectrum peak) with the twin cavity as having a band gap of 1.5 eV. If the twin cavity were a true (meta-)stable state, we might expect to see a feature in the absorption spectrum about 1.1 eV redshifted from the peak; however, this would be at an excitation energy lower than the experimental spectrum turn-on at 0.8 eV. Of course, one cannot directly compare band gaps to the absorption spectrum, but simulating the optical absorption spectrum should be straightforward, and would provide an additional important validation of the simulation results against experiment.
- The (metastable) twin cavity structure should be particularly observable in the Polarized Transient Hole Burning (PTHB) spectrum, the observation and persistence of which reports on the anisotropy of the solvent shell.[2] The twin cavity in this sense represents an extreme anisotropy that would be expected to yield a persistent polarized bleach signal to the red of the absorption max. However, no persistent PTHB signal of the hydrated electron is observed experimentally.[1] The formalism for computing PTHB observables is presented in Ref. [2]: only equilibrium ground-state dynamics is needed, so the authors' trajectories could straightforwardly be used as input after computing electronic transition energies and dipoles.

e. The simulation cell used is rather small (47 waters) that finite-size effects are likely substantial. The authors don't report the simulation cell size, but 47 neutral waters at ambient density would have a cell length of 11.3Å, and given the small solvation volume of the hydrated electron, its cell size should be similar. The concern of finite-size effects is exacerbated for the twin cavity: given solvation shell radii of $\sim 3\text{Å}$ (see Fig. 1i), the first solvation shells of the twin cavity structure span essentially the entire simulation shell ($4 \times 3\text{Å} = 12\text{Å}$) and under periodic-boundary conditions, the two cavities interact with their neighboring periodic images as strongly as the primary image. This might provide some artificial stability of the twin cavity structure.

2) The authors don't make clear whether they verified their ML model using structures that the model was not explicitly trained to (e.g. by performing Cross Validation). The authors state in the SI that in training they used "100 additional representative configurations from quantum molecular dynamics for the bulk water" and "600 additional structures" for the solvated electron. To my eye, there appear to be around 100 PIMD data points in Fig S1(a), and around 600 PIMD data points in Fig. S1(b), suggesting that the authors indeed tested their ML model on the underlying training set. This then raises a serious concern that their ML model might suffer from overfitting and/or selection bias, and be unable to accurately describe the potential energy surface away from the training points. To show the validity of the model, the authors should test the ML energy and force errors against MP2 data on structures randomly selected outside of the training set. Of particular importance will be showing that energies & forces are equally well reproduced for the following types of hydrated electron structure they observe: cavity, twin cavity, delocalized.

3) Another curiosity related to the training is seen in Fig. S1(b): two PIMD energies appear as outliers away from the rest of the PIMD data (approximately 50 eV lower in total potential energy compared to the next lowest PIMD total energy). Their energies are close to the AIMD data – is this a simple mislabelling error, or are they really from the PIMD simulation? If the latter, can the authors provide a reason for their low potential energy?

4) The authors state that "The mean $\langle e\text{-O} \rangle$ coordination number is 4.5, which corresponds to the experimentally deduced value $\langle e\text{-O} \rangle = 23$ ". Actually, the review article of Ref. 23 does not report an "experimentally deduced value". The only experimental determination of the coordination number of the hydrated electron is a six-coordinate model based on ESR experiments on alkaline, glassy water at $T=77\text{K}$. [3] The relevance of this result to ambient water at $\text{pH}=7$ has long been questioned [4,5]. The authors should not claim agreement with experiment for this observable.

5) Radial Distribution Functions (RDF) are reported in Fig. 1h-i. It's not clear whether these included configurations corresponding to twin cavity structures – their lack of overall spherical symmetry about a single center makes an RDF inappropriate. Likewise, for the classical simulation, it's not clear whether the reported delocalized electron structures were included in the RDF. Again, their inclusion might be inappropriate since the electron centroid may be ill defined for a delocalized state. The authors should clarify. But with that said, it is interesting to note that the e-H RDFs (both quantum and classical) have non-zero values at distances of less than 0.5Å. This puts the oxygen atom (occasionally) within 1.5Å of the electron, which is less than the Van der Waals radius of the oxygen atom. This is consistent with the recent fluxional cavity picture put forward in Ref. 5 and it would be helpful for this connection to be pointed out.

[1] Cavanagh; Schwartz, Chem. Phys. Lett. 396, 359 (2004)

[2] Shkrob, Chem. Phys. Lett. 467, 84 (2008)

[3] Kevan, Acc. Chem. Res. 14, 138 (1981)

[4] Turi; Borgis, J. Chem. Phys. 117, 6186 (2002)

[5] Glover; Schwartz, J. Chem. Theory Comput. 16, 1263 (2020)

Reviewer #3 (Remarks to the Author):

In this paper, the authors use their previous MP2 simulations of the hydrated electron to train a neural network, allowing construction of an effective potential that can be used in much larger simulations of the hydrated electron as well as the incorporation of nuclear quantum effects for the water protons. This is a highly ambitious work, clearly the results of lots of intellectual and computational effort, and ultimately needs to appear in the literature somewhere. The problem is, in its current form, there's so much going on here that I felt that some of the big picture story gets lost, particularly the connection to experiment. If the authors can clarify some of these details and keep the thread better supported, I'd be happy to consider the appropriateness of a revision for publication in Nature Comm.

First, there's the BPNN potential itself. The authors show the energies from the ML and MP2 calculations, but other than this don't comment on the nature of the potential. In particular, I worry that a lot of the 'double cavity' that the authors see are a direct result of the periodic boundary conditions used in the training set. Although the training set by itself is an impressive set of calculations, with only 47 waters, the simulation cell is barely twice as big as the electron's gyration diameter, which had me worried about the authors' original Angew. Chem. paper on the subject. This means that the inclusion of (spurious) finite-size effects in the NN potential could lead to all sorts of unexpected behaviors as the system size is expanded. Did the authors ever try smaller simulations for training the NN potential (since bigger simulations aren't computationally feasible)? Or DFT-based based simulations at similar sizes? I'd really like to see how the NN potential changes with different sized training sets are used. Maybe the authors could retrain the NN potential on the 25 or 30 closest waters to the electron from the original configurations and see how that affects the resulting potential and then the behavior in the larger simulation?

Next, it would be really interesting to take configurations generated with the NN potential and see how different the results would be for the Turi-Borgis pseudopotential run in those same configurations, as this is the potential most widely used in the literature. There's so much more information in the NN potential, based on MP2, than the T-B potential based on Hartree-Fock, that it would really help the reader to understand what that extra information is doing as far as the electron is concerned.

I'm also still not clear about the authors' incorporation of quantum effects for the protons. It appears that the NN potential was trained on simulations that don't include dispersion of the protons, and then the protons were treated by PIMD in the LAMMPS simulation using the NN potential. Given that proton dispersion affects both the water H-bonding and potentially the structure of the solvated electron, doesn't this mean that the levels of theory are mismatched? Given that the NN potential doesn't really know that the protons are quantum, I'm very surprised at how much difference the addition of PIMD made to the electron's structure and dynamics. I did read through the SI, and it's possible I missed it, but the authors really need to clarify the consistency between the NN potential and the use of PIMD after-the-fact.

In Figure 1, the authors show the base results of the simulations. It's clear from Figs. 1b/c that the inclusion of nuclear quantum effects makes a huge difference, presumably from the presence of the 'double-cavity' trajectories. The double-cavity is such an extraordinary claim that it needs extraordinary evidence to support it. The simplest evidence would be the UV-Visible absorption spectrum. My glance at Fig. 1b suggests that the snapshots with small radius of gyration and small energy gap would lead to features in the absorption that likely would not mesh with experiment. I understand that the spectrum is not readily available from the simulations, but there are many things the authors could do to approximate it, e.g., using TD-DFT to estimate the excited-state energy gap

and transition dipoles on a handful of uncorrelated configurations. The authors already have run DFT calculations to visualize the spin density, so this would not be an unreasonable amount of work. Or even more simply, approximating the absorption spectrum by binning the energy gaps and assuming the Condon approximation, just to get a rough sense of the peak position and width. If the authors could show that inclusion of nuclear quantum effects actually brought the calculated spectrum into better agreement with experiment, I'd be jumping up and down about publishing this work. As-is, I'm skeptical. The authors could also comment on whether their calculated spectrum is homogeneously broadened, as shown by experiment (my guess is that the separate populations of the double cavity and single cavity configurations would be inhomogeneously broadened based on the time scale over which they persist in Fig. 1a, and this would be contrary to experiment).

Since the potential is trained on a 47-water simulation, it should be 47x3 dimensional (if it's been reduced in dimensionality somehow, this is not at all clear from the text or SI). Since the NN simulation has far more waters/dimensionality than 47, what keeps different uncorrelated regions of the larger simulation cell from each trying to display their own hydrated electron behavior, particularly given the finite-size issues mentioned above? Could something like this be responsible for the double-cavity (two independent regions of the simulation each trying to create their own electron, with some 'interference' between them)? Did the authors try running the NN potential with nuclear quantum effects using only the original 47 waters to see if the double-cavity still appears? At the moment, I'm just not convinced that the double cavity is real, and this is most of the story of this paper.

The authors comment on the number of H-bonds made to the electron. Can they examine the structure in terms of the simple model proposed by Kumar et al. (JPC A, 119, 9148, 2015), i.e. tetrahedral cavity, by looking at angular or other distributions of the first-shell waters?

It is gratifying to see that the $g(r)$'s show a clear cavity, as snapshots alone are insufficient to shed light on the cavity/non-cavity controversy. But how spherically-symmetric is the electron on average? The authors give the radius of gyration but not the asphericity (which they must have calculated as per the SI). Does $g(r)$ even make sense for the double-cavity structure? What fraction of the configurations have this structure vs. the normal cavity structure? Could the authors characterize the cavity and its fluctuations using a counting coordinate, similar to that done by Schwarts (JCTC 16, 1263, 2020)? Please give the reader something more concrete to take home concerning the electron's size and shape!

The idea that the double-cavity configurations could contribute to the 'anomalously' high diffusion constant of the electron is really interesting; given this, I'm surprised that the incorporation of PIMD didn't change the calculated diffusion constant within error. Can the authors explain why this is? Does the other cavity-walking mechanism become less probable when PIMD is added? Or is this simply because the NN potential doesn't know about the PIMD, as mentioned above?

Finally, I didn't find the vibrational analysis to be terribly convincing. In theory, the NN potential knows about how local polarizability of the water affects the O-H vibrational frequency, but whether this information is properly transferred to the classical (or PIMD) proton dynamics through their VAC is unclear to me, particularly given the fact that the simulation appears not to reproduce the experimental frequencies, either for pure water or the electron. Really this needs to be a normal mode analysis, and it's unclear how simply taking the VAC weights/ignores the normal modes that would be displaced upon excitation, which is what the resonance Raman experiment actually measures. The authors show only the centroid frequency and not the full line width/shape for the experiments for bulk water, which would be a helpful comparison, as would overlaying the Tauber/Mathies experimental Raman spectrum of the electron. The fact that the first-shell water vibrations are redshifted is at least encouraging, but without a better match to experiment or a better justification of

this way of estimating the vibrational spectrum, it seems like this might be better relegated to the SI.

Overall, this is a highly ambitious work on a subject of great interest to the broad Nature Comm. community. Even with all the benchmarking in the SI, however, I'm unconvinced that the NN potential has really captured the essence of the 'ghost' due to finite size effects in the training set, or that the addition of PIMD for the protons was done consistently with the construction of the NN potential. I don't understand why adding dispersion to the protons should create a double-cavity, and the double cavity seems too persistent to be connected to the experimental object. The authors make little attempt to make a direct connection to experiment (e.g., VDE's/photoelectron spectra could be determined using the same DFT calculations used to visualize the electron's spin density, and I already mentioned estimating the absorption spectrum above, etc.) other than the vibrational spectrum, which only qualitatively connects to experiment at best. If the authors can address these concerns, I'd be happy to review this paper again, but in its present form, it feels like the authors have bitten off more than they can chew in this paper as far as publishing in Nature Chem. is concerned.

Reply to Reviewers for the submitted manuscript: "Simulating the Ghost: Quantum Dynamics of the Solvated Electron"

October 6, 2020

We would like to thank the Editor and the three Reviewers for taking into consideration our work and suggesting improvements to the manuscript. In what follows we provide our answers to the reviewers' comments and indicate in detail all the changes implemented into the main manuscript and into the supplementary material.

Reviewer #1

Issues for revision:

(1) I think that the discussion of DFT vs. MP2 for liquid water in the introduction is a bit disingenuous. I understand that a common trope in the introduction to high-profile papers is basically to say "the state of the art has terrible problems", but that doesn't mean this is right, and the authors are pushing it a bit. For example, they say "neat water is already challenging for DFT" and cite a Perspective by Michaelides (Ref. 9) titled "How good is DFT for water?", but if you actually *read* that paper, the answer that Michaelides offers is actually: pretty good. There are some issues, for sure, but the situation is certainly much improved as compared to the early days of *ab initio* MD and anyway there are issues with the MP2 description of liquid water as well, which are not acknowledged here (Hirata, 10.1021/acs.jpcclett.5b02430; Vandevondele, 10.1021/jz401931f). Specifically, MP2 water is too dense and probably has a too-high boiling temperature. Both issues are likely related to the fact that MP2 significantly overestimates dispersion interactions, and the problems would be much more severe in a liquid where the dispersion effects were more significant. (They are certainly collectively significant in liquid water but amount to only a few percent of the binding energy of water dimer.) The present work can stand on its own without needing to misrepresent the current state-of-the-art. MP2 is a good level of theory (if one can afford it) for this particular system but that doesn't mean that we should throw out DFT, and it doesn't mean that MP2 is a good choice for everything even if it is feasible.

Generally, we agree that MP2 is not the best method for all systems and properties (especially when the computational price is considered). However, we believe there is enough evidence that MP2 is a very accurate approach for bulk aqueous systems. There might be some issues for MP2 water as suggested by Hirata and co-workers, *e.g.* high density and boiling temperature. However, that work is based on the fragment MP2 method using double-zeta basis set in a small unit cell (32 water molecules). In contrast, our current work relies on the canonical implementation of Del Ben et al. Its application to bulk liquid water with triple-zeta basis (ref 10; and Del Ben et al, J. Chem. Phys. 2015 143, 054506, 2015) has not shown any major deficiencies. In particular, the density at ambient conditions is only 2% higher than the experimental one. In fact, we have used the forces from these studies to train our NN potential for neat water. In addition, MP2 has been shown to be in excellent agreement to CCSD(T) for two- and three-body interaction according to Paesani's critical assessment [J. Chem. Theory Comput. 2013, 9, 11031114].

MP2 is known to overestimate dispersion interaction. This is, however, detrimental for aromatic and non-polar systems. For hydrogen-bonded systems such deficiencies are not known to us (see also work of Del Ben et al on ice, Del Ben et al, J. Phys. Chem. Lett. 2014, 5, 4122).

We also agree that a proper choice of DFT functional can provide satisfactory results for water dimer, trimer and clusters against CCSD(T) benchmarks, and when used in molecular dynamics give a reasonable structure of liquid water. DFT is the most successful approach in electronic structure theory and trying to scorn it would be ridiculous.

Nevertheless, we show that MP2 is at least a very promising method for bulk aqueous systems and as a purely *ab initio* method it relieves us from being involved in a seemingly endless discussion of the best functional

for water. We believe we should be given the right to express our opinion, based on solid data. In any case, our machine learning potentials for both liquid water and solvated electron will be made public so that it would be possible to address potential issues.

(2) Speaking of things that are disingenuous, the authors make comments about how DFT cannot be trusted for radicals due to self-interaction (not wrong but overstated)... and then they proceed to visualize all the data using spin densities computed with hybrid DFT. They use a functional that Ambrosio and Pasquarello have previously used to examine electronic energy levels of both neat liquid water and of e-(aq), and I think this functional is fine for the purpose for which it is used here. That said, it was still rather jarring to segue directly from criticism of DFT in the introduction to repeated use of DFT later in the paper.

Indeed, self-interaction can be compensated by introducing certain fraction of exact exchange. PBEh(α) with 40% of exact exchange (following Pasquarello's work) exhibits good agreement with MP2 in spin density distribution as shown in the SI and is thus a reasonable visualizer. However, one should not forget that the frames are generated from ML-MP2 MD, and not by PBEh(α). The differences in structures resulting from long-time MD can be significant. For instance, the gyration radii reported by Pasquarello are larger than those found in this work, although spin density distributions have been computed by the same functional.

We would like to emphasize again that DFT as a huge family of methods is extremely flexible. Proper choice of parameters including empirical tuning may provide accurate results for a particular system (as in case of liquid water). The functional of Pasquarello and coworkers is the example of such an approach. Although it is obviously useful and applied here, it lacks the power of generalization. Purely *ab initio* methods avoid this, thus, we try to stick to them as much as possible.

(3) On pg. 6, the authors note that their simulated coordination number of 4.5 "corresponds to the experimentally deduced value and is lower than the best DFT estimate of 4.7". If you put uncertainties on those values representing fluctuations, my guess is they are the same number so I think it's a cheap shot to say MP2 agrees with experiment but DFT does not. Furthermore, what is the experiment? Those numbers seem about right for various room-temperature simulations but the only experimental work of which I'm aware that gets at coordination number are ESR experiments at liquid nitrogen temperatures, which is not relevant here. It's possible I've missed some literature but it seems like the 4.5 number is likely an indirect inference at best.

Indeed, it is an indirect inference, therefore, the statement is made less strict.

(4) Regarding the vibrational spectra, it is compelling that the authors recover a splitting in the HOD bend, which is a feature that Ben Schwartz's non-cavity model never recovered even as he was pushing the resonance Raman spectrum as the thing that proved the veracity of his non-cavity interpretation. (This cognitive dissonance was noted in Ref. 30.) However, I was surprised by the lack of agreement with experiment in the O-H stretching region, although the simulations do pick up on the broadening on the low-energy side of the OH feature, comparing neat water to e-(aq), which is a hallmark of this system. Do the authors have an explanation for the lack of quantitative agreement in the stretching region? What is the state of anharmonicity in these simulations? The classical calculations I would expect to be overly harmonic and that may account for how this feature appears at higher wavenumbers in the classical simulation, but I don't have a good understanding of why the errors should be so large once nuclear quantum effects are introduced, at what ought to be a good level of electronic structure theory.

To achieve excellent quantitative agreement with experimental vibrational spectra is non-trivial, as it requires both high-precision electronic structure theory and quantum dynamics methods. Both of them are still open questions and under discussion and development. In particular, no PIMD flavour is recognized as a golden standard providing uncompromising accuracy for most systems and properties. In fact, one can solve analytically the transition frequency for a Morse Oscillator (parametrized to model an O-H bond) and compare with different quantum dynamical methods. As seen in the Fig.1 (From our recent paper [J. Chem. Phys. 152, 124104 (2020)]), the TRPMD simulation overestimates the O-H vibrational frequency compared to the analytical solution. The mismatch at stretching region is attributed that TRPMD shows an artificial broadening [J. Phys. Chem. Lett. 2017, 8, 15451551]. Thus, this effect is a clear deficiency of the method, which is still one of the best in the market. On the other hand, our simulation is able to reproduce the experimental frequencies at bending region. And the downshift from the O-H stretching can be fully captured by our simulations.

Figure 1: Vibrational density of states as calculated by the velocity autocorrelation $C_v v(\omega)$ for a Morse oscillator parametrized to model an O–H bond. The dashed black line shows the $0 \rightarrow 1$ transition frequency.

Technical issues:

(1) A more standard notation for the density functional that is used would be PBEh(alpha) rather than PBE(alpha), where the “h” indicates that it is a hybrid.

The notation of PBEh(alpha) has been used in the SI.

(2) In the simulations of electron injection (Fig. 1a), is the system truly zero-gap at $t=0$? That would not surprise me, as the electron is probably fully delocalized or at least spread over a number of disjoint pre-existing traps or voids. My question is: how do the authors perform MP2 calculations for a zero-gap system? I realize this is a question mainly about Ref. 13 but I did not review that paper so I am asking it now.

When the $t=0$, there is still a gap around 1 eV according to our hybrid calculations. To perform MP2 calculations for a zero-gap system is impossible since the MP2 comes from perturbation theory. In the Ref.13, the G_0W_0 band gap at $t=0$ is still non-zero. Hartree-Fock gap, relevant for MP2, is around 1 eV as well. That is large enough to avoid divergence of the perturbation theory correction.

(3) Pg. 7: I do not understand the sentence “These transient diffusion events... are associated with a characteristic structural and electronic signature in Fig. 2(f)”. What are the signatures to which the authors refer?

The transient diffusion events are associated to the twin-cavity intermediate, which has particular geometric structure and spectroscopic signature.

(4) There are some capitalization issues with the references that probably go back to the Bibtex files the authors are using.

The references have been updated

Reviewer #2

(1) The twin cavity structure observed in PIMD simulations of the hydrated electron is highly surprising. First, this is not merely a rare-event fluctuation: Fig. 1(a) shows many trajectories that visit low band-gap structures characteristic of the twin-cavity state, and they persists for at least 7 ps in at least one trajectory, suggesting the twin cavity is a (meta-)stable state of the hydrated electron in the authors' model. The authors should clarify the percentage of time a twin cavity structure is observed. Nevertheless, I have strong doubts this corresponds to the physical reality of the hydrated electron. Instead, I suspect it is a simulation/ML artefact for the following reasons:

a. One would expect the formation of two separate cavities and reduction of charge density in each cavity by half compared to a single cavity to be strongly disfavored in a high dielectric solvent like water. It is hard to imagine any physical driving force to the formation of the twin cavity.

As a matter of fact, the charge would be distributed between the cavities if the latter were symmetric and static. In our case the cavities are never equivalent. Therefore, the charge oscillates between them without filling both for an extended time. What is shown in the figure is a moment of transfer in the transient diffusion. Such structures are short-lived. We find this in agreement with the reviewer's argument about high dielectric constant of water. Thus, the twin void is observed, but electron prefers staying in one of its halves. The physical reason for opening the second cavity is nuclear quantum effects which disrupt hydrogen bond network more intensively than is observed in the classical dynamics.

b. The BPNN forcefield underlying the PIMD simulations may have issues of overfitting/selection bias that are discussed in more detail in point 2) below.

Figure 2: A comparison between the Machine learning energies computed using lammmps, and the energies computed using UMP2 for solvated electron. From left to right are energy validation including 100, 200, 600 PIMD data and additional 100 twin-cavity data. The additional twin-cavity structure are highlighted in green crosses.

The selection of dataset is based on the atomic fingerprints method [J. Chem. Phys. 148, 241730 (2018)]. The first machine learning potential has been trained based on the previous data. Based on the first machine learning potential, we generate many molecular dynamics trajectories (About 60,000 snapshots) from classical and quantum dynamics. We then choose 100 representative snapshots and recalculate with MP2. The machine learning potential has been gradually improved by generating more points based on machine learning potential, selecting representative snapshots and recalculating at the MP2 level of theory then adding to the datasets. Cross validation has been performed at each iteration.

In the last plot of the Figure 2, we cross-validated 100 additional twin-cavity points (marked in green crosses). The quality of our machine learning potential turned out to be reliable though a few twin-cavity energies from the centroid are slightly underestimated by our MLP. However, RPMD, in principle, won't be able to see those points as the beads are corresponding to the datasets from PIMD datasets.

Furthermore, we trained a new machine learning potential which included more twin-cavity datasets. We can still see the transient diffusion mechanism as proposed.

c. The (metastable) twin cavity structure should be detectable in the optical absorption spectrum. The experimental absorption spectrum is a featureless single peak at 1.7 eV.[1] Although the authors did not compute the optical spectrum, they report the average band gap at 2.6 eV (notably higher than the experimental absorption spectrum peak) with the twin cavity as having a band gap of 1.5 eV. If the twin cavity were a true (meta-)stable state, we might expect to see a feature in the absorption spectrum about 1.1 eV redshifted from the peak; however, this would be at an excitation energy lower than the experimental spectrum turn-on at 0.8 eV. Of course, one cannot directly compare band gaps to the absorption spectrum, but simulating the optical absorption spectrum should be straightforward, and would provide an additional important validation of the simulation results against experiment.

Among our 40 trajectories, the twin-cavity structure has been observed in few trajectories and most of the trajectories still hold the single-cavity picture. As correctly pointed out by the reviewer, band gaps are not equivalent to optical transition frequencies. Evaluating the absorption is, however, less straightforward than it may seem. One will have to use TDDFT, which is very sensitive to the amount of the exact exchange in the functional. Our PBEh(α) with 40% of exact exchange proven good as a "proxy" for MP2 spin densities may not be the best choice for the spectrum. Using different amounts of exact exchange one will predict very different absorption spectra and in the absence of the reference any judgement will be unmotivated.

In any case, we have undertaken an attempt to compute TDDFT spectrum hybrid DFT using Quantum Espresso. It has proved to be too expensive for our goal as computing exact exchange in the plane wave basis is non-trivial and for a system with 141 a massively parallel implementation is needed, which is not available. Although this is being under development in CP2K, the implementation is not currently validated and optimized.

Performing non-empirical calculation with CC2, ACD(2) could solve the issue, but those methods to the best of our knowledge are not implemented for periodic systems.

In fact, to align the absolute level of solvated electron from ab-initio calculations is fully non-trivial as demonstrated by Gali [Nat Commun 9, 247 (2018)] and Pasquarello [J. Phys. Chem. Lett. 2017, 8, 9, 2055–2059]. Naively, we can re-align our band-gap in term of experimental absorption spectrum, assuming there is a linear relation between computational band-gap and experimental excitation energy.

d. The (metastable) twin cavity structure should be particularly observable in the Polarized Transient Hole Burning (PTHB) spectrum, the observation and persistence of which reports on the anisotropy of the solvent shell.[2] The twin cavity in this sense represents an extreme anisotropy that would be expected to yield a persistent polarized bleach signal to the red of the absorption max. However, no persistent PTHB signal of the hydrated electron is observed experimentally.[1] The formalism for computing PTHB observables is presented in Ref. [2]: only equilibrium ground-state dynamics is needed, so the authors' trajectories could straightforwardly be used as input after computing electronic transition energies and dipoles.

We have included a more detailed discussion of twin- and single-cavities in both the main text and the SI. It turns out that, although the twin-void is observed it is likely that only one half it is filled with spin density most of the time of its existence. Instead of occupied both halves, the electron shuffles between them. MP2 spin densities for twin-cavity are much less anisotropic, which explains the lack of persistent signal in PTHB.

As soon as PTHB is concerned, formalism is known for one-particle model and its application for solvated electron was worth a separate publication (I. A. Schkrob, Chem. Phys. Lett. 467, 84 (2008)). Its *ab initio* extension and implementation in the *ab initio* context under periodic boundary conditions is highly non-trivial, e.g. calculation of transition dipoles. As method developers we do not find this task realistic within the revision, although will consider it for the future research.

e. The simulation cell used is rather small (47 waters) that finite-size effects are likely substantial. The authors don't report the simulation cell size, but 47 neutral waters at ambient density would have a cell length of 11.3Å, and given the small solvation volume of the hydrated electron, its cell size should be similar. The concern of finite-size effects is exacerbated for the twin cavity: given solvation shell radii of 3Å (see Fig. 1i), the first solvation shells of the twin cavity structure span essentially the entire simulation shell ($4 \times 3 \text{ Å} = 12 \text{ Å}$) and under periodic-boundary conditions, the two cavities interact with their neighboring periodic images as strongly as the primary image. This might provide some artificial stability of the twin cavity structure.

It was an error not to put the simulation box size in the manuscript. The simulation is carried out in a cubic cell with a cell length of 11.295 Å.

2) The authors don't make clear whether they verified their ML model using structures that the model was not explicitly trained to (e.g. by performing Cross Validation). The authors state in the SI that in training they used "100 additional representative configurations from quantum molecular dynamics for the bulk water" and "600 additional structures" for the solvated electron. To my eye, there appear to be around 100 PIMD data points in Fig S1(a), and around 600 PIMD data points in Fig. S1(b), suggesting that the authors indeed tested their ML model on the underlying training set. This then raises a serious concern that their ML model might suffer from overfitting and/or selection bias, and be unable to accurately describe the potential energy surface away from the training points. To show the validity of the model, the authors should test the ML energy and force errors against MP2 data on structures randomly selected outside of the training set. Of particular importance will be showing that energies forces are equally well reproduced for the following types of hydrated electron structure they observe: cavity, twin cavity, delocalized.

Please, see above.

3) Another curiosity related to the training is seen in Fig. S1(b): two PIMD energies appear as outliers away from the rest of the PIMD data (approximately 50 eV lower in total potential energy compared to the next lowest PIMD total energy). Their energies are close to the AIMD data – is this a simple mislabelling error, or are they really from the PIMD simulation? If the latter, can the authors provide a reason for their low potential energy?

Those are indeed from the PIMD simulations. The first initial PIMD started from a snapshot which was sampled using classical molecular dynamics. The first few PIMD configurations are very close to those from classical molecular dynamics. Therefore, the energy is much closer to that of classical structures.

4) The authors state that "The mean e—O coordination number is 4.5, which corresponds to the experimentally deduced value 23". Actually, the review article of Ref. 23 does not report an "experimentally deduced value". The only experimental determination of the coordination number of the hydrated electron is a six-coordinate model based on ESR experiments on alkaline, glassy water at T=77K.[3] The relevance of this result to ambient water at pH=7 has long been questioned[4,5]. The authors should not claim agreement with experiment for this observable.

We agree and the claim has been removed.

Reviewer #3

Before addressing specific points, we would like to make a few general remarks.

First, our model is in the end a classical force field (FF), although in the form of neural network potential. The very fact that an FF can provide meaningful results for a non-classical system is one of the main results of the work (surprising even to us). As other FFs, ours provides access to path-integral molecular dynamics due to computational efficiency, but at a price of being oblivious to electronic properties. Hence, we find it consistent to concentrate on the explicit properties of nuclei: structures, vibration frequencies, diffusion and alchemy exchange energies. To the best of our knowledge, the latter are reported for the first time. The frequencies and diffusion coefficients are reported only in few works, thus, we hope to have contributed to a less charted area, delivering observable quantities - in direct contact with the experiment. It is, of course, tempting to probe nuclear quantum effects on electronic properties. However, we do it mainly for visualization purposes as we believe that doing this properly would require methods similarly complicated as MP2 (GW, BSE etc.) and, ideally, their combination with machine learning. This may be an excellent task for a follow-up work. In addition, even a brief analysis of later publications shows that original contributions almost never focus on more than two properties (structures and binding energies [ref. [22]], structures and electronic spectra (ref. [6, 16]), structures and resonance Raman spectra [ref. [31] etc.).

Second, the results reported here are obtained at the limit of computational power of Europe's largest supercomputer facility, the Swiss National Supercomputing Centre. We would be happy to probe the size-dependence of NQE at MP2 level or at least with hybrid DFT and check the double-cavity observation. However, that is barely possible at all (MP2) and impossible within three months given for revision (hybrid DFT). To be specific, transient diffusion is a rare event, therefore, one would need at least 50 PIMD simulations of 10 ps length, each including at least 10 beads. It should also be done for two system sizes (say 64 and 128 molecules). Compare this with the work of Pasquarello's group (ref. [23]): they performed 40 classical (one bead) trajectories, 1 ps long each, for two systems sizes: 64 and 128 water molecules. This required 100 less CPU resources than what we would need and yet ref. [23] is based on state-of-the-art calculations.

Let us now address specific points:

First, there's the BPNN potential itself. The authors show the energies from the ML and MP2 calculations, but other than this don't comment on the nature of the potential. In particular, I worry that a lot of the 'double cavity' that the authors see are a direct result of the periodic boundary conditions used in the training set. Although the training set by itself is an impressive set of calculations, with only 47 waters, the simulation cell is barely twice as big as the electron's gyration diameter, which had me worried about the authors' original Angew. Chem. paper on the subject. This means that the inclusion of (spurious) finite-size effects in the NN potential could lead to all sorts of unexpected behaviors as the system size is expanded. Did the authors ever try smaller simulations for training the NN potential (since bigger simulations aren't computationally feasible)? Or DFT-based based simulations at similar sizes? I'd really like to see how the NN potential changes with different sized training sets are used. Maybe the authors could retrain the NN potential on the 25 or 30 closest waters to the electron from the original configurations and see how that affects the resulting potential and then the behavior in the larger simulation?

To retrain the NN potential on 25 or 30 water molecules may be not feasible due to the high cost of MP2 calculations even for these sizes. The generation of data set needs enormous computational resources (over a million hybrid XC50 Cray node hours) and unlikely to yield a meaningful outcome. The choice of 47 water models have been discussed in the original Angew. Chem. paper and again condenses to computational resources. MP2 scales as N^5 and performing the simulations even in a cell with 64 water molecules would go beyond computational reach of world's largest supercomputers.

Next, it would be really interesting to take configurations generated with the NN potential and see how different the results would be for the Turi-Borgis pseudopotential run in those same configurations, as this is the potential most widely used in the literature. There's so much more information in the NN potential, based on MP2, than the T-B potential based on Hartree-Fock, that it would really help the reader to understand what that extra information is doing as far as the electron is concerned.

We agree that a comparison with different one-electron pseudopotential models, such as Turi-Borgis pseudopotential, will provide very useful knowledge to understand how MP2 improve the description of the solvated electron. However, we were not able to find the implementation of the Turi-Borgis pseudopotential in publicly available software. Thus, the comparison would require implementation and evaluation in addition to the simulations, which is not achievable within the revision deadline. In addition, a detailed comparison with T-B

potential may be beyond the scope of our current work. We believe the potential research collaborations should be done with Turi's group to understand the difference between T-B potential and our NN potential.

I'm also still not clear about the authors' incorporation of quantum effects for the protons. It appears that the NN potential was trained on simulations that don't include dispersion of the protons, and then the protons were treated by PIMD in the LAMMPS simulation using the NN potential. Given that proton dispersion affects both the water H-bonding and potentially the structure of the solvated electron, doesn't this mean that the levels of theory are mismatched? Given that the NN potential doesn't really know that the protons are quantum, I'm very surprised at how much difference the addition of PIMD made to the electron's structure and dynamics. I did read through the SI, and it's possible I missed it, but the authors really need to clarify the consistency between the NN potential and the use of PIMD after-the-fact.

Nuclear quantum effects have been simulated using PIMD methods. PIMD models quantum motion of nuclei within the Born-Oppenheimer approximation. The regions of the phase space sampled by quantum dynamics are slightly different from the classically sampling regions (normally they are also larger). Therefore, we append our "classical" dataset with snapshots taken from PIMD. Thus, the final training dataset includes the phase space accessed by both classical and quantum dynamics, and our NN potential is able to handle the energetics and forces which are needed to propagate molecular dynamics simulations.

In Figure 1, the authors show the base results of the simulations. It's clear from Figs. 1b/c that the inclusion of nuclear quantum effects makes a huge difference, presumably from the presence of the 'double-cavity' trajectories. The double-cavity is such an extraordinary claim that it needs extraordinary evidence to support it. The simplest evidence would be the UV-Visible absorption spectrum. My glance at Fig. 1b suggests that the snapshots with small radius of gyration and small energy gap would lead to features in the absorption that likely would not mesh with experiment. I understand that the spectrum is not readily available from the simulations, but there are many things the authors could do to approximate it, e.g., using TD-DFT to estimate the excited-state energy gap and transition dipoles on a handful of uncorrelated configurations. The authors already have run DFT calculations to visualize the spin density, so this would not be an unreasonable amount of work. Or even more simply, approximating the absorption spectrum by binning the energy gaps and assuming the Condon approximation, just to get a rough sense of the peak position and width. If the authors could show that inclusion of nuclear quantum effects actually brought the calculated spectrum into better agreement with experiment, I'd be jumping up and down about publishing this work. As-is, I'm skeptical. The authors could also comment on whether their calculated spectrum is homogeneously broadened, as shown by experiment (my guess is that the separate populations of the double cavity and single cavity configurations would be inhomogeneously broadened based on the time scale over which they persist in Fig. 1a, and this would be contrary to experiment).

We have undertaken an attempt to compute TDDFT spectrum hybrid DFT using Quantum Espresso. It has proved to be too expensive for our goal as computing exact exchange in the plane wave basis is non-trivial and for a system with 141 a massively parallel implementation is needed, which is not available. Although this is being under development in CP2K, the implementation is not currently validated and optimized.

Since the potential is trained on a 47-water simulation, it should be 47x3 dimensional (if it's been reduced in dimensionality somehow, this is not at all clear from the text or SI). Since the NN simulation has far more waters/dimensionality than 47, what keeps different uncorrelated regions of the larger simulation cell from each trying to display their own hydrated electron behavior, particularly given the finite-size issues mentioned above? Could something like this be responsible for the double-cavity (two independent regions of the simulation each trying to create their own electron, with some 'interference' between them)? Did the authors try running the NN potential with nuclear quantum effects using only the original 47 waters to see if the double-cavity still appears? At the moment, I'm just not convinced that the double cavity is real, and this is most of the story of this paper.

The training and the production runs are performed for the same system, including 47 water molecules. The reason for this is that scaling the system size in the production run will also scale the number of solvated electrons: having 94 water molecules in the cell will correspond to two excess electrons.

The double-cavity only appears when we include NQEs in our simulations, which is a strong evidence, that it is not a size effect.

The authors comment on the number of H-bonds made to the electron. Can they examine the structure in terms of the simple model proposed by Kumar et al. (JPC A, 119, 9148, 2015), i.e. tetrahedral cavity, by looking at angular or other distributions of the first-shell waters?

This can be done, however, we prefer the PAMM analysis as our major tool for rationalizing geometric structures. In this case, the tetrahedral/non-tetrahedral cavity shape is synonymous to coordination number. Since the coordination number is 4.5 half of the cavities are tetrahedral (4-coordinated), half are bi-pyramidal (5-coordinated). NQE have little effect on the coordination number and consequently on the cavity shapes. Thus, we would refrain from performing this type of the analysis, although all the trajectories will be published, so that any interested researcher will be able to process them.

It is gratifying to see that the $g(r)$'s show a clear cavity, as snapshots alone are insufficient to shed light on the cavity/non-cavity controversy. But how spherically-symmetric is the electron on average? The authors give the radius of gyration but not the asphericity (which they must have calculated as per the SI). Does $g(r)$ even make sense for the double-cavity structure? What fraction of the configurations have this structure vs. the normal cavity structure? Could the authors characterize the cavity and its fluctuations using a counting coordinate, similar to that done by Schwartz (JCTC 16, 1263, 2020)? Please give the reader something more concrete to take home concerning the electron's size and shape!

We indeed calculated the anisotropy of the solvated electron which is about 0.0467 ± 0.0397 . The $g(r)$ does not fully make sense for the double-cavity structure. However, double-cavity structures make minor contributions to the trajectory-averaged RDF(s).

We did provide the coordination number to the solvated electron in Fig1.(b,c). Instead using a continuous Fermi function for the counting function, we simply count the coordination of hydrogen to electron with a cutoff of 2.5\AA , which is within the first water shell of solvated electron. The take home message is similar to that done by Schwarz (JCTC 16, 1263, 2020)

The idea that the double-cavity configurations could contribute to the 'anomalously' high diffusion constant of the electron is really interesting; given this, I'm surprised that the incorporation of PIMD didn't change the calculated diffusion constant within error. Can the authors explain why this is? Does the other cavity-walking mechanism become less probable when PIMD is added? Or is this simply because the NN potential does not know about the PIMD, as mentioned above?

We agree that the transient diffusion involved a double-cavity intermediate contribute the "anomalously" high diffusion constant of the electron. When the PIMD is used, a slightly higher diffusion coefficients has been estimated despite the fact that the diffusion coefficients calculated from classical and quantum dynamics are within their statistical uncertainty of each other. When the PIMD is used, only the transient diffusion mechanism have been observed and non-transient diffusion is similar to the classical molecular dynamics The NN potential can fully handle the PIMD simulations as mentioned before.

Finally, I didn't find the vibrational analysis to be terribly convincing. In theory, the NN potential knows about how local polarizability of the water affects the O-H vibrational frequency, but whether this information is properly transferred to the classical (or PIMD) proton dynamics through their VAC is unclear to me, particularly given the fact that the simulation appears not to reproduce the experimental frequencies, either for pure water or the electron. Really this needs to be a normal mode analysis, and it's unclear how simply taking the VAC weights/ignores the normal modes that would be displaced upon excitation, which is what the resonance Raman experiment actually measures. The authors show only the centroid frequency and not the full line width/shape for the experiments for bulk water, which would be a helpful comparison, as would overlaying the Tauber/Mathies experimental Raman spectrum of the electron. The fact that the first-shell water vibrations are redshifted is at least encouraging, but without a better match to experiment or a better justification of this way of estimating the vibrational spectrum, it seems like this might be better relegated to the SI.

This point has also been addressed at the point of Reviewer # 1

Sincerely,

Jinggang Lan, Venkat Kapil, Piero Gasparotto, Michele Ceriotti, Marcella Iannuzzi, Vladimir V. Rybkin

REVIEWER COMMENTS

Reviewer #2 (Remarks to the Author):

In the revised manuscript, the authors have addressed some of my concerns, but the main issue that I have with the paper remains unaddressed: the authors propose a radical new picture of the hydrated electron (occasional dual cavity structure) without any demonstration of how such a structure is compatible with known experimental observables, namely the absorption spectrum exhibiting a single featureless peak, and the lack of persistent Polarized Transient Hole Burning (PTHB) spectrum. From the authors' rebuttal it appears that they misunderstood the nature of the anisotropy that PTHB measures (more details below), and so they don't seem to have appreciated the importance of this observable. I want to give the authors one more chance to address the electronic spectroscopy (and I note that I was not the only reviewer to request it), but if they insist on not computing absorption and PTHB spectra [even though the former has been done at the TDDFT level over 10 years ago: *J. Am. Chem. Soc.* **132**, 10000 (2010)], then I cannot recommend the manuscript for publication in *Nature Communications*. I am still unconvinced the dual cavity is a physical attribute of the real hydrated electron and not a finite size artefact from their small simulation cell (that shows up only in PIMD due to quantum effects weakening hydrogen bonds). I need to see more evidence to convince me otherwise. With that said, the idea of machine learning MP2 trajectories to get an atomic-only representation of the total potential for a solvated excess electron is ground-breaking, and their classical MD results are a really nice demonstration of this. I would suggest resubmitting the manuscript with a focus on the classical MD to a more specialized journal such as *Phys. Rev. Lett.*

Detailed comments:

1) The authors have seemingly misunderstood the nature of anisotropy that PTHB measures – it is not simply anisotropy in the ground-state wavefunction, but rather anisotropy in the electronic excitations, i.e. whether the transition dipoles for excitations in the pump energy window persist in a certain orientation to then be detected as a bleach signal in a time-delayed polarized probe pulse. Thus it matters not that their electronic ground state has low anisotropy and primarily stays in one of the two dual-cavities. For dual cavity configurations, the first electronic excitation would surely transfer the electron between cavities (hence the shuttling of the electron between cavities due to thermal fluctuations) and the lowest energy transition dipole would then align along the dual-cavity axis and have an orientation that persists as long as the dual-cavity persists. In other words, as I stated in my previous review: the dual cavity should give rise to a persistent PTHB signal, in stark disagreement with experimental observation.

2) The authors state that “energies of a few twin-cavity structures from the centroid are slightly underestimated by our MLP”. But unless they have labelled the axes of SI Fig 2 incorrectly, their data shows the opposite: the energies of the twin-cavity structures are *overestimated* by MLP.

3) The authors never responded to my previous point 5), which was missing in their rebuttal: Radial Distribution Functions (RDF) are reported in Fig. 1h-i. It's not clear whether these included configurations corresponding to twin cavity structures – their lack of overall spherical symmetry about a single center makes an RDF inappropriate. Likewise, for the classical simulation, it's not clear whether the reported delocalized electron structures were included in the RDF. Again, their inclusion might be inappropriate since the electron centroid may be ill defined for a delocalized state. The authors should clarify. But with that said, it is interesting to note that the e-H RDFs (both quantum and classical) have non-zero values at distances of less than 0.5Å. This puts the oxygen atom (occasionally) within 1.5Å of the electron, which is less than the Van der Waals radius of the oxygen atom. This is consistent with the recent fluxional cavity picture put forward in *J. Chem. Theory*

Comput. **16**, 1263 (2020) and it would be helpful for this connection to be pointed out.

Reviewer #3 (Remarks to the Author):

Overall, while I really wanted to like this paper, I felt like the authors did almost nothing to address my comments from the previous round. With almost all my concerns still outstanding, I unfortunately cannot recommend this paper for publication in Nature Comm.

First, the authors seemed to mis-understand my suggestion about the size dependence. I wasn't suggesting they re-run the original MP2 trajectories at different sizes, but instead to use the existing trajectories and retrain the NN potential at different sizes. For example, instead of training the NN potential on all 47 waters in the MP2 trajectory, one could use only the 30 closest waters to the electron's center of mass, and then run the NN simulations to see how different the final result turns out to be. Since the electron sits in a (single) cavity in these simulations, the closest 30 or so waters out of 47 should still constitute 2 solvation shells, but the difference between ~ 30 and 47 would really highlight whether or not the double cavity behavior was coming from the NN potential trying to act independently in different regions of the larger classical simulations.

I was also disappointed on the authors response to training the TB potential. Yes, this is not available in commercially-available software, but I recently had a first-year graduate student code this from scratch in only a few weeks, and an experienced graduate student would probably need even less time. It's also possible that Turi, Herbert, Schwartz and others who have explored the behavior of this potential have put configurations up on the web that could be used for the training set. Simply saying "this would be hard because its not in the code I own" is not an excuse for not doing the science needed to understand exactly what's happening in the simulated system.

I still didn't understand the response about whether the PIMD trajectories are part of the NN training set or not. The text implies that only the raw MP2 trajectory with classical nuclei was used to train the NN potential. The response suggests the NN potential was trained on a mix of classical and PIMD-treated nuclei. If the latter is correct, did the authors run PIMD-MP2 dynamics? I had thought the PIMD was added only to the NN simulations? The distinction is important because the authors claim to see large differences in behavior with PIMD vs. classical nuclei, but if the NN potential doesn't 'know' about PIMD nuclei, how can it predict the correct behavior? And if it does 'know', how was that information included? None of this has been clarified by the authors' response.

I was again dissatisfied with the response to my questions about whether the absorption spectrum was consistent with experiment, or not. The answer is "we can't do this in commercially available code". One could use a less expensive functional for the TD-DFT, or even get a rough estimate by using the K-S orbitals from the DFT calculations used to visualize the spin density. One of the other referees also commented on this point -- the presence of the double-cavity states, which is the key result of this work, is likely not consistent with experiment.

How can NQEs not affect the local bonding and structure of the electron if they sometimes induce a double-cavity? The authors' statement to this effect makes no sense given the data in the paper that shows a reasonably significant fraction of double-cavity states along the trajectory!

I agree that the $g(r)$ doesn't make sense for double-cavity structures, which is why I suggested other means to quantify the electron's shape. The anisotropy of 0.047 is pretty spherical -- does this number include the double-cavity structures or not?

Again, I don't understand statements like "the diffusion coefficient with and without NQEs were the same within error, but higher when NQEs are included". Either the statistics are sufficient to say one is higher than the other, or they're not. The authors can't have it both ways!

The response to my comments about the Raman spectrum, also raised by one of the other referees, wasn't satisfactory. The agreement with experiment is poor enough that I felt that this information should be moved to the SI. The authors admit that TRMPD overestimates the stretch frequency and that this is a "clear deficiency of the method". The bending frequency does look better, but without discussion about the isotope effect (which is much more important than the tiny downshift), I still don't find this data convincing.

Overall, what the authors have tried is ambitious -- they've taken the most state-of-art ab initio simulation of the hydrated electron done to date and used it to train an NN potential to explore NQEs for this problem. But, the findings have some extraordinary claims -- particularly the double cavity -- and it's not entirely clear why the double cavity arises. Does the same thing happen for a non-quantum electron with a similar structure, like TB, where the NN potential could be trained on a larger system? Is the presence of the double cavity states consistent with the experimental absorption spectrum and also transient hole-burning experiments? Without answers for questions like these, I cannot believe that the observed behavior is the result of a weird finite-size interaction between the NN potential and the implementation of PIMD, and this is why I reluctantly conclude that the work is not acceptable for publication in Nature Comm.

Reply to Reviewers for the submitted manuscript: "Simulating the Ghost: Quantum Dynamics of the Solvated Electron"

November 24, 2020

We would like to thank the Editor and the three Reviewers for taking into consideration our work and suggesting improvements to the manuscript. In what follows we provide our answers to the reviewers' comments and indicate in detail all the changes implemented into the main manuscript and into the supplementary material.

Reviewer #2

1) The authors have seemingly misunderstood the nature of anisotropy that PTHB measures – it is not simply anisotropy in the ground-state wavefunction, but rather anisotropy in the electronic excitations, i.e. whether the transition dipoles for excitations in the pump energy window persist in a certain orientation to then be detected as a bleach signal in a time-delayed polarized probe pulse. Thus it matters not that their electronic ground state has low anisotropy and primarily stays in one of the two dual-cavities. For dual cavity configurations, the first electronic excitation would surely transfer the electron between cavities (hence the shuttling of the electron between cavities due to thermal fluctuations) and the lowest energy transition dipole would then align along the dual-cavity axis and have an orientation that persists as long as the dual-cavity persists. In other words, as I stated in my previous review: the dual cavity should give rise to a persistent PTHB signal, in stark disagreement with experimental observation.

In response to the comments of the reviewer we have carried out TDDFT simulations of both single- and double-cavities with the hybrid PBEh(50) functional, producing spin density distributions similar to those of MP2 for both cavity types. Including the first, i.e. the most important, excited state for a single cavity produces a wide peak with a maximum at ca. 2.0 eV, the largest amplitude corresponding to the p-type orbital. This proves that the system size is adequate and the methods are reasonable. The spectrum of the double-cavity is also a broad continuous distribution with a maximum at ca. 0.9 eV. As correctly anticipated by the reviewer, the first excited state is the charge transfer between the two cavities for structures. This must indeed cause bleaching of the ground state.

Nevertheless, we do not find these observations in direct contradiction to the experiment. First, double-cavity structures are observed only less than 10% of our simulations (3 out of 40). This is at the edge of sensitivity of pump-probe experiments, as ground-state bleaching is a very fine effect. Likewise, conventional electronic absorption spectroscopy would not trace minor admixtures of double-cavity signals. Second, transient spectroscopy measurements known to us have been designed to find the bleaching of the ground state due to p-state quasi-degeneracy in a classical single cavity. For that, excitation pulses with the wave length of around 700 nm (from 500 to 800 nm) have been applied. We predict the double cavity to absorb at longer wave length: ca. 1400 nm.

The corresponding discussion is now added to the main text and as well as the section on electronic spectroscopy in the Supplementary Information.

2) The authors state that "energies of a few twin-cavity structures from the centroid are slightly underestimated by our MLP". But unless they have labelled the axes of SI Fig 2 incorrectly, their data shows the opposite: the energies of the twin-cavity structures are overestimated by MLP.

We thank the referee for pointing out this error. The axes were indeed incorrectly labeled, and we have corrected this mistake in the SI.

3) The authors never responded to my previous point 5), which was missing in their rebuttal: Radial Distribution Functions (RDF) are reported in Fig. 1h-i. It's not clear whether these included configurations corresponding to twin cavity structures – their lack of overall spherical symmetry about a single center makes an RDF inappropriate. Likewise, for the classical simulation, it's not clear whether the reported delocalized electron structures were included in the RDF. Again, their inclusion might be inappropriate since the electron centroid may be ill defined

for a delocalized state. The authors should clarify. But with that said, it is interesting to note that the e-H RDFs (both quantum and classical) have non-zero values at distances of less than 0.5Å. This puts the oxygen atom (occasionally) within 1.5Å of the electron, which is less than the Van der Waals radius of the oxygen atom. This is consistent with the recent fluxional cavity picture put forward in *J. Chem. Theory Comput.* 16, 1263 (2020) and it would be helpful for this connection to be pointed out.

We haven't include the double cavity structures in the RDF since the gyration centers of such structure doesn't make sense due to the lack of spherical symmetry as suggested by the referee. Similarly the delocalized electron structures haven't been included either due to the same reasons. We have revised our manuscript accordingly to avoid potential misunderstanding. We are grateful for the relevant reference. We have added a brief discussion of how quantum delocalization reduces entropic pressure.

Reviewer #3

1) First, the authors seemed to misunderstand my suggestion about the size dependence. I wasn't suggesting they re-run the original MP2 trajectories at different sizes, but instead to use the existing trajectories and retrain the NN potential at different sizes. For example, instead of training the NN potential on all 47 waters in the MP2 trajectory, one could use only the 30 closest waters to the electron's center of mass, and then run the NN simulations to see how different the final result turns out to be. Since the electron sits in a (single) cavity in these simulations, the closest 30 or so waters out of 47 should still constitute 2 solvation shells, but the difference between 30 and 47 would really highlight whether or not the double cavity behavior was coming from the NN potential trying to act independently in different regions of the larger classical simulations.

We appreciate the further explanation of the simulations the reviewer would like to see. If we understand correctly, they suggest taking existing (47-water molecules) MP2 simulations, and training the simulations based only on the configurations of the 30 water molecules that are closest to the center of mass of the excess electron. Unfortunately, this approach is not practicable; while it is true that forces are defined for individual water molecules, and so one could restrict the training to only use those closest to the electron, the total energy is a necessary piece of information, and is only defined for the entirety of the 47-molecules-plus-electron system. Furthermore, the forces acting on the 30 water molecules closest to the electron are also determined by the positions of the water molecules that are further away from it. Unless we misunderstood again the reviewer's request, a test of the effect of changing the system size cannot be achieved without rerunning the entire training protocol, which would be far too demanding, and unlikely to bring insights that justify the computational effort.

We added a sentence explaining clearly why the potential is restricted to using precisely the same number of water molecules that it was trained on

Note that, contrary to what is usually the case, our machine-learning potential should not be ran for a different system size than it was trained for. This is because the presence of an excess electron is an additional parameter of the reference calculations, and so the potential is effectively trained at a fixed concentration of electrons. Changing system size without further training would be equivalent to introducing additional electrons, which may lead to nonphysical results.

In fact, a scenario like the one described by the reviewer, where two cavities open because of the NN acting independently on different parts of the simulation box, might happen if we were to simulate a box that is two times as large. There are several reasons we can bring to convince the reviewer that nothing of the sort is happening here: 1. the cutoff of the NN descriptors are large enough to go beyond the first shell of neighbors, so the model knows that having an independent localization event is not possible; 2. the mechanism of cavity formation is a transient event starting from an existing cavity, ruling out the possibility that this happens independently in two parts of the simulation box that are both similar to the initial, neat water state of the reference calculations; 3. a significant amount of double-cavity structures have been included in the training set, and indeed the energy of some out-of-sample validation structures does not show large deviations from the NN predictions. The energetics of the double cavities predicted by the NN is compatible with that predicted by MP2 calculations.

2) I was also disappointed on the authors response to training the TB potential. Yes, this is not available in commercially-available software, but I recently had a first-year graduate student code this from scratch in only a few weeks, and an experienced graduate student would probably need even less time. It's also possible that Turi, Herbert, Schwartz and others who have explored the behavior of this potential have put configurations up on the web that could be used for the training set. Simply saying "this would be hard because its not in the code I own" is not an excuse for not doing the science needed to understand exactly what's happening in the simulated system.

We respectfully beg to disagree. Repeating simulations with a TB model is extremely unlikely to provide additional insights. If the simulations gave similar results to ours, it might provide weak additional confirmation that the mechanism is robust, but would not be conclusive - as one could blame artifacts due to the limitation of the semiempirical model. If the simulations gave different results, there would be no reason to trust TB over our NN model - on the contrary, given the extensive validation of the NN against MP2 references, we would be inclined to consider the discrepancy a sign of the failure of the TB model.

In particular, the TB model can't be used to cross-check the most striking finding of our work: the double-cavity and transient diffusion. Indeed, the suggested TB potential is designed to describe the standard single-cavity, as the electron is kept close to its fixed center by the confining potential. Even if quantum dynamics leads to the opening of the second void the electron would not be able to migrate there. Moreover, it has been suggested that since the TB is based on the Hartree-Fock theory, comparing it to our ML-MP2 method would reveal the effect of electron correlation. In fact, in the TB potential only the water-electron interactions are based on Hartree-Fock theory, whereas water-water interactions are outsourced to an empirical force field. Thus,

there is no clear physical interpretation of potential differences between the TB and ML-MP2-based simulations.

If the authors of the TB model had made it openly and readily available, in the spirit of open science promoted by this Journal, it would have made sense to run a quick test to satisfy the curiosity of the reviewer. Since that is not the case, and embarking in a new implementation would require disproportionate effort and risks leading to results that are affected by coding errors, we stand by our choice not to perform these additional calculations. Ultimately, our simulations provide semi-quantitative accuracy against a number of experimental probes, which is, in our opinion, a vastly more stringent test than the comparison between two approximate methodologies.

3) I still didn't understand the response about whether the PIMD trajectories are part of the NN training set or not. The text implies that only the raw MP2 trajectory with classical nuclei was used to train the NN potential. The response suggests the NN potential was trained on a mix of classical and PIMD-treated nuclei. If the latter is correct, did the authors run PIMD-MP2 dynamics? I had thought the PIMD was added only to the NN simulations? The distinction is important because the authors claim to see large differences in behavior with PIMD vs. classical nuclei, but if the NN potential doesn't 'know' about PIMD nuclei, how can it predict the correct behavior? And if it does 'know', how was that information included? None of this has been clarified by the authors' response.

We first train a machine learning potential based on the original MP2 data. Based on this first machine learning potential, we run PIMD and generate a quantum dataset. Those PIMD trajectories are further added into the training set. Thus, we extend our training set and improve the quality of our machine learning potential. The classical trajectory indeed does not know the PIMD nuclei and extrapolation is needed in the first version of machine learning potential – although the trajectory remains stable. We would also like to point out that for neat water, training on classically generated trajectories, leads to a robust enough model to perform stable PIMD which samples physically relevant regions of phase space. So this is not the first instance a potential trained on classical trajectories is used to further generate a PIMD sample configurations [doi:10.1063/5.0016004]. As expected, when we further included the sampled PIMD data, the new machine learning potential was found to describe accurately the regions of phase space associated with quantum fluctuations. This set was iteratively updated until self-consistency, *i.e.* until PIMD simulations ceased to generate structures dissimilar to those included in the training set.

4) I was again dissatisfied with the response to my questions about whether the absorption spectrum was consistent with experiment, or not. The answer is "we can't do this in commercially available code". One could use a less expensive functional for the TD-DFT, or even get a rough estimate by using the K-S orbitals from the DFT calculations used to visualize the spin density. One of the other referees also commented on this point – the presence of the double-cavity states, which is the key result of this work, is likely not consistent with experiment.

First of all, we would like to point out, that Kohn-Sham band gaps have already been reported as early as in the original manuscript. In any case, given the importance of this point to address the validity of the double-cavity structure, we have performed TDDFT simulations of absorption spectra of single- and double-cavities using the fresh developer version of CP2K. We want to stress that our reluctance to embark in these calculations is due to the fact that these are far from being routine calculations: previously published results on solvated electron (by Herbert and Jungwirth) used cluster models. We usually refrain from using code that is not publicly released, precisely to avoid putting other researchers in the condition of being asked to run calculations that are not available in open source, academic codes. The corresponding section has been added to the SI, showing the spectra and the orbitals involved in the first electronic transition. The spectrum of the single cavity is consistent with the experiments and previous calculations: a broad peak with the maximum at ca. 2.0 eV (experimental value is 1.7 eV), corresponding to the s- p-type orbital transitions. The spectrum of the double cavity is also a broad peak centered around 0.9 eV, corresponding to the s-s-type orbital transition between the cavities. Keeping in mind the low concentration of the double cavity, smaller intensities and the absence of sharp features in the double-cavity spectrum we conclude that its existence does not contradict experimental observations. At the same time pump-probe experiments at lower frequencies (ca. 0.9 eV) could be used to probe the existence of twin-cavities.

5) The response to my comments about the Raman spectrum, also raised by one of the other referees, wasn't satisfactory. The agreement with experiment is poor enough that I felt that this information should be moved to the SI. The authors admit that TRMPD overestimates the stretch frequency and that this is a "clear deficiency of the method". The bending frequency does look better, but without discussion about the isotope effect (which is much more important than the tiny downshift), I still don't find this data convincing.

There is no exact quantum nuclear dynamics technique that is applicable to a problem of this complexity, and effectively TRPMD is the most accurate approximation that is at all affordable in this case. For the gas phase molecule described by the spectroscopically accurate Partridge-Schwenke model, it is well known that TRPMD blue shifts the stretching frequency by around 60 cm^{-1} [doi:10.1063/1.5100587], which is similar to

the discrepancy observed in our results. On the other hand, the bending mode of a water molecule is known to be accurately described by TRPMD [doi:10.1063/1.5100587]. The closely-related (and not necessarily more accurate) CMD technique would be approximately an order of magnitude more demanding, because of the need to introduce adiabatic decoupling between centroid and ring polymer modes. The errors of TRPMD and of all approximate quantum dynamics techniques increase with the quantum mechanical nature of the modes being investigated, which explains the well-known difficulties in the stretching region, which we had already commented on. There is another reason for a larger error in the stretch region than in the bend one, well known from molecular quantum chemistry. Modest errors in harmonic force constants lead to larger discrepancies in the high-frequency modes than in lower-frequency ones. This is often coped with by scaling the force field by a constant scaling factor (although more complex schemes exist). Most methods including MP2, HF and GGA DFT overestimate stretching frequencies. Of course, it is known that MP2 is not the "gold standard" of quantum chemistry. However, just as TRPMD it is the best available for the type of systems we study.

When it comes to the bending region, experience and previous studies suggest that both the MP2 reference, and the TRPMD VDOS, should be much more reliable. Indeed, our results show a very substantial improvement relative to those of Herbert and coworkers, the best available at the moment. The stretching frequencies for H₂O are compatible: ca. 3100 cm⁻¹. The bending frequencies are summarized in the table:

	exp. (Tauber)	Herbert	this work
H ₂ O	1609	1850	1599
D ₂ O	1175	1350	1186
HOD	1399/1465	1520/1600	1338/1399

Thus, we have reached qualitative accuracy for normal and heavy water. For semi-heavy water, the discrepancy between theory and experiment decreased by 50%, and the value of the bending mode splitting is quantitatively correct. We regard this as a major merit of our work, which deserves to appear in the main text.

When it comes to isotope effects, a comprehensive discussion has already been presented by Herbert and coworkers, so we only discussed them briefly in terms of HB populations, leaving a more detailed discussion for the supplementary. A novelty of our work, which we discuss in more detail, is the first explicit calculation of equilibrium isotope exchange based on quantum alchemical calculations, that show that transfer of the electron from normal to heavy water is unfavourable, which adds to our knowledge solvated electron in isotopic mixtures, and is compatible with some indirect experimental evidence, as discussed.

6) How can NQEs not affect the local bonding and structure of the electron if they sometimes induce a double-cavity? The authors' statement to this effect makes no sense given the data in the paper that shows a reasonably significant fraction of double-cavity states along the trajectory!

The fraction of frames corresponding to a double cavity structure is less than 10%. We have shown in the first revision (see supplementary section "Twin-cavity") that the electron most of the time occupies one of the twin-cavity halves and shuffles between them, instead of occupying both at the same time. Structures in which the electron is truly delocalized, such as that shown in Fig. 1, follow the transient diffusion and are very rare. When the electron occupies one of the two cavities, its "size" and spin density distribution are very similar those in the single cavity (Fig. 4 in the supplementary). One may also compare the s-type SOMOs (mostly responsible for spin density distribution) for single and double cavities shown in Figure 8 in the supplementary. They are very similar, with larger differences appearing for the excited states – which however have no bearing on structural properties.

7) Again, I don't understand statements like "the diffusion coefficient with and without NQEs were the same within error, but higher when NQEs are included". Either the statistics are sufficient to say one is higher than the other, or they're not. The authors can't have it both ways!

The reviewer is correct that we were overinterpreting the difference between the classical and quantum value. Statistics is not sufficient to make a conclusive statement. We have amended the discussion of diffusion that currently reads

The diffusion coefficient obtained by quantum dynamics ($0.40 \pm 0.03 \text{ \AA}^2/\text{ps}$) is within the statistical error bar of the classical value ($0.36 \pm 0.04 \text{ \AA}^2/\text{ps}$), both numbers in reasonable agreement with the experimental measurements of $0.475 \pm 0.048 \text{ \AA}^2/\text{ps}$.

8) I agree that the $g(\mathbf{r})$ doesn't make sense for double-cavity structures, which is why I suggested other means to quantify the electron's shape. The anisotropy of 0.047 is pretty spherical – does this number include the double-cavity structures or not?

The number includes double-cavities. We remind, however, that those represent less than 10% of the structures. Even in simulations where double-cavities are present, the electron is most of the time localized in

one of the two subcavities and shuffles between them instead of occupying both (this is shown in Fig. 1, but this is a rare structure illustrating the event of transient diffusion).

Sincerely,

Jingang Lan, Venkat Kapil, Piero Gasparotto, Michele Ceriotti, Marcella Iannuzzi, Vladimir V. Rybkin

REVIEWERS' COMMENTS

Reviewer #2 (Remarks to the Author):

The latest revision to the manuscript addresses my previous concerns, and I think it is now suitable for publication in Nature Communications subject to the following changes below.

1) On P9, last paragraph, the authors state "The computed absorption maximum for the single-cavity structures, corresponding to the s-type to p-type orbital transition, is located at ca. 2 eV (Fig. 7 and 8 in the SI) in a reasonable agreement with the experiment (1.7 eV)." However, a detailed reading of the SI reveals that the authors only show the absorption spectrum arising from the s-like ground to first p-like excited state. It is well understood that the absorption spectral peak of the hydrated electron at 1.7 eV arises from three overlapping s-to-p sub-bands. If only one excited state is used to compute the spectrum then it should peak to the red of 1.7 eV and be narrower than the experimental width (FWHM ~ 0.82 eV, see *Chem. Phys. Lett.* **438**, 234 (2007)); however, the author's computed single-state single-cavity spectrum peaks at ~ 2 eV, with a FWHM of ~ 1.25 eV. Therefore, the authors should not directly compare their computed ca. 2 eV peak to the experimental 1.7 eV peak, but instead compute the spectrum including at least three excited states and simply acknowledge that their spectrum is significantly blue shifted and broadened compared to experiment. The decomposition of the spectrum into p sub-bands is also important to predict the Transient Hole Burning spectroscopy.

2) On P9, last paragraph, the authors state "We expect them to give a signal in transient bleaching experiments³¹, although the search should be done at lower frequencies and can be hindered by the method's sensitivity to the low-concentration twin-cavity structures." A second hindrance is that the expected bleach signal due to the dual cavity at ~ 0.9 eV will likely be obscured by the excited-state absorption of the electron in the same region, see e.g. *J. Phys. Chem.* **98**, 3450 (1994).

Reviewer #3 (Remarks to the Author):

Overall, I'm still not convinced that the double cavity, which is the key result of this work, is not an artifact. The authors have attempted to calculate the contribution of these configurations to the absorption spectrum, and indeed they should have a contribution that appears around 1400 nm. Yet, the red side of the electron's experimental absorption spectrum fits very nicely to the standard Gaussian-Lorentzian form; there's no room in what's been experimentally observed for the electron to have even weak absorption transitions corresponding to charge-transfer states $\sim 10\%$ of the time (which is a lot in my opinion). If the authors really believe that what they've found is real, they should calculate the pTHB experiment exciting this transition at 1400 nm and make a prediction for what would be found experimentally. As-is, the authors are claiming to have found this extraordinary feature, but then they also claim that it has no experimentally-testable manifestations. This type of conclusion is highly unsatisfying, to say the least.

The authors also claim that most of the time the double cavity does exist, the electron only occupies one of the two cavities. Given this, why does the second cavity exist at all? There is a huge entropic penalty to move the water out of a region of space and leave that space empty. How does a nearby electron restructure the water to permit this to happen in $\sim 10\%$ of all configurations? I can see that adding NQE's can weaken the local H-bonding, potentially making the second cavity easier to form. But if weakened H-bonds were really the cause of the second cavity, presumably the double cavity would not exist in D2O, which has nuclei that behave essentially classically. I'm not aware of any

qualitative differences in the behavior the hydrated electron in H₂O and D₂O (other than a small spectral shift that is consistent with the stronger H bonding in D₂O, making the spectrum in D₂O behave as if it were effectively at a temperature about 15 K lower than the corresponding spectrum in H₂O); in particular, the electron still has enhanced diffusion in D₂O, which is one of the key arguments the authors give to justify the presence of the double-cavity. The same Gauss-Lorentzian form fits the spectrum in both H₂O and D₂O, and there are no features on the red side of the spectrum in H₂O that don't exist in D₂O. Thus, without further testing on other models such as TB, I'm left with the reluctant conclusion that the existence of the double cavity is an artifact of the ML model.

Reply to Reviewers for the submitted manuscript: "Simulating the Ghost: Quantum Dynamics of the Solvated Electron"

December 9, 2020

We would like to thank the Editor and the Reviewers for taking into consideration our work and suggesting improvements to the manuscript. In what follows we provide our answers to the reviewers' comments and indicate in detail all the changes implemented into the main manuscript and into the supplementary material.

Reviewer #2

On P9, last paragraph, the authors state "The computed absorption maximum for the single-cavity structures, corresponding to the s-type to p-type orbital transition, is located at ca. 2 eV (Fig. 7 and 8 in the SI) in a reasonable agreement with the experiment (1.7 eV)." However, a detailed reading of the SI reveals that the authors only show the absorption spectrum arising from the s-like ground to first p-like excited state. It is well understood that the absorption spectral peak of the hydrated electron at 1.7 eV arises from three overlapping s-to-p sub-bands. If only one excited state is used to compute the spectrum then it should peak to the red of 1.7 eV and be narrower than the experimental width (FWHM 0.82 eV, see Chem. Phys. Lett. 438, 234 (2007); however, the author's computed single-state single-cavity spectrum peaks at 2eV, with a FWHM of 1.25 eV. Therefore, the authors should not directly compare their computed ca. 2 eV peak to the experimental 1.7 eV peak, but instead compute the spectrum including at least three excited states and simply acknowledge that their spectrum is significantly blue shifted and broadened compared to experiment. The decomposition of the spectrum into p sub-bands is also important to predict the Transient Hole Burning spectroscopy.

We have recomputed the spectrum of the single cavity taking all p-states in account. Unfortunately, not for every structure it was possible to identify all three of them, which is likely the consequence of the small simulation cell and the known TDDFT drawback of generating spurious charge transfer states. Qualitatively, the results have not changed. The procedure is now described in the SI.

On P9, last paragraph, the authors state "We expect them to give a signal in transient bleaching experiments³¹, although the search should be done at lower frequencies and can be hindered by the method's sensitivity to the low-concentration twin-cavity structures." A second hindrance is that the expected bleach signal due to the dual cavity at 0.9 eV will likely be obscured by the excited-state absorption of the electron in the same region, see e.g. J. Phys. Chem. 98, 3450 (1994).

We have included this observation and reference in the main text.

Reviewer #3

Overall, I'm still not convinced that the double cavity, which is the key result of this work, is not an artifact. The authors have attempted to calculate the contribution of these configurations to the absorption spectrum, and indeed they should have a contribution that appears around 1400 nm. Yet, the red side of the electron's experimental absorption spectrum fits very nicely to the standard Gaussian-Lorentzian form; there's no room in what's been experimentally observed for the electron to have even weak absorption transitions corresponding to charge-transfer states 10% of the time (which is a lot in my opinion). If the authors really believe that what they've found is real, they should calculate the pTHB experiment exciting this transition at 1400 nm and make a prediction for what would be found experimentally. As-is, the authors are claiming to have found this extraordinary feature, but then they also claim that it has no experimentally-testable manifestations. This type of conclusion is highly unsatisfying, to say the least. The authors also claim that most of the time the double cavity does exist, the electron only occupies one of the two cavities. Given this, why does the second cavity exist at all? There is a huge

entropic penalty to move the water out of a region of space and leave that space empty. How does a nearby electron restructure the water to permit this to happen in 10% of all configurations? I can see that adding NQE's can weaken the local H-bonding, potentially making the second cavity easier to form. But if weakened H-bonds were really the cause of the second cavity, presumably the double cavity would not exist in D2O, which has nuclei that behave essentially classically. I'm not aware of any qualitative differences in the behavior the hydrated electron in H2O and D2O (other than a small spectral shift that is consistent with the stronger H bonding in D2O, making the spectrum in D2O behave as if it were effectively at a temperature about 15 K lower than the corresponding spectrum in H2O); in particular, the electron still has enhanced diffusion in D2O, which is one of the key arguments the authors give to justify the presence of the double-cavity. The same Gauss-Lorentzian form fits the spectrum in both H2O and D2O, and there are no features on the red side of the spectrum in H2O that don't exist in D2O. Thus, without further testing on other models such as TB, I'm left with the reluctant conclusion that the existence of the double cavity is an artifact of the ML model.

We believe that our data on double-cavity is already a prediction for PTHB experiments: such predictions are most of the time made based on absorption spectrum and qualitative arguments such as orbital symmetry and degeneracy. In fact, reviewer 2, who was the first to bring PTHB spectroscopy in the discussion, appears to be satisfied with the presented analysis.

As a matter of fact, transient diffusion (and double cavities) are not observed in our quantum simulations of heavy water, making it clearly a manifestation of NQEs. This had not been made clear in the text. We have corrected this by adding an unequivocal statement.

We would also like to highlight that 10% is the upper limit of double-cavity concentration (3 simulations out of 40). Moreover, these 3 trajectories encompass not only double-cavity structures, but also single cavity ones. Therefore, the "shoulder" in the red part of the spectrum produced by the double cavity shown in Figure 7 of the SI must be even less pronounced.

We are afraid, we never report or discuss enhanced diffusion in D2O. Moreover, keeping in mind significant statistical errors we have refrained to claim that transient mechanism enhances diffusion at all. We find this plausible but our data do now allow us to make a definite conclusion.

We respect the reviewer's opinion on the TB model. We believe it is difficult to add anything to the discussion at this point. We could only reiterate that in our opinion the only sane theoretical test would be actual *ab initio* PIMD not based on simple models and machine-learning potentials.

As an independent evidence for transient diffusion, we attach a video depicting our not published simulation of solvated electron at the surface of water. This is classical *ab initio* MD based on hybrid DFT. In this simulation, we observe transient diffusion close to the surface. Near-surface layers can be viewed as macroscopic defects in hydrogen bond network. Such defects allow for transient diffusion and double-cavity formation without NQEs. In the bulk, NQEs create defects, as anticipated by the reviewer, and allow for transient diffusion, which is much less common than at the surface.

Finally, we would not view our predictions as impossible to validate experimentally. Double-cavity is a marginal effect which requires fine experimental measurements (keeping in mind that even a fundamental property of the solvated electron as binding energy was accurately measured only recently, Luckhaus et al. Science, 3(4), e1603224). On the other hand, rapid progress in experimental techniques must make it possible in the near future.

Sincerely,

Jinggang Lan, Venkat Kapil, Piero Gasparotto, Michele Ceriotti, Marcella Iannuzzi, Vladimir V. Rybkin